# Amyloid formation and depolymerization of tumor suppressor p16[INK4a] are regulated by a thiol-dependent redox mechanism

Sarah G. Heath [1], Shelby G. Gray [2], Emilie M. Hamzah[2], Karina M. O'Connor[1], Stephanie M. Bozonet [1], Alex D. Botha [1], Pierre de Cordovez[1], Nicholas J. Magon[1], Jennifer D. Naughton[1], Dylan L. W. Goldsmith[2], Abigail J. Schwartfeger [2], Margaret Sunde [3], Alexander K. Buell [4], Vanessa K. Morris [2,5] ✉ & Christoph Göbl [1,5] ✉

The conversion of a soluble protein into polymeric amyloid structures is a process that is poorly understood. Here, we describe a fully redox-regulated amyloid system in which cysteine oxidation of the tumor suppressor protein p16[INK4a] leads to rapid amyloid formation. We identify a partially-structured disulfide-bonded dimeric intermediate species that subsequently assembles into fibrils. The stable amyloid structures disassemble when the disulfide bond is reduced. p16[INK4a] is frequently mutated in cancers and is considered highly vulnerable to single-point mutations. We find that multiple cancer-related mutations show increased amyloid formation propensity whereas mutations stabilizing the fold prevent transition into amyloid. The complex transition into amyloids and their structural stability is therefore strictly governed by redox reactions and a single regulatory disulfide bond.

Amyloid fibrils are protein structures associated with a range of proteopathic diseases, as well as a variety of biological functions in different organisms[1,2]. Amyloids are characterized by their fibrillar polymeric state, where β-sheet secondary-structural motifs run perpendicular to the long fibril axis[3]. Detailed studies into the mechanisms of amyloid formation have contributed to the understanding of amyloids in health and disease, for example by highlighting the roles of intermediate oligomeric species in toxic processes, and by elucidating potential avenues for therapeutic intervention[4,5]. Therefore, the mechanisms of amyloid formation are of central interest for a better understanding of this class of shape-shifting proteins. Although tremendous progress has been made in the understanding of the structural transition of a number of amyloid proteins, detailed molecular insights into the misfolding process are difficult to obtain. In particular the onset of formation and details about intermediate species have proven elusive[6].

We recently discovered that in the presence of oxidants, the all-α helical protein p16 forms β-sheet based amyloid fibrils[7]. p16 (also referred to as p16[INK4a]) is a tumor suppressor protein that is frequently mutated in cancer[8,9]. It is an ankyrin-repeat type protein that inhibits cyclin-dependent kinases 4 and 6 (CDK4/6) to regulate cell division[10]. Its solution structure has been determined by nuclear magnetic resonance previously (Fig. 1a), as well as structures from co-crystallization with its major interaction partner CDK6[9,11]. Despite its compact fold, p16 has been noted to have an unusually low energy barrier of unfolding, which was speculated to be a functional property[12]. The delicate stability was further confirmed by analysis of various types of tumors, where large numbers of single-point mutations were identified that lead to loss-of-function and uncontrolled cell division[9,13].

Here, we report a reversible, redox-regulated fibril formation mechanism. p16 undergoes aggregation into amyloid upon oxidation of a single cysteine residue by physiological oxidants. The amyloid

[1]Mātai Hāora - Centre for Redox Biology and Medicine, Department of Pathology and Biomedical Science, University of Otago, Christchurch, New Zealand. [2]School of Biological Sciences, University of Canterbury, Christchurch, New Zealand. [3]School of Medical Sciences and Sydney Nano, The University of Sydney, Sydney, Australia. [4]Department of Biotechnology and Biomedicine, Technical University of Denmark, 2800 Lyngby, Denmark. [5]Biomolecular Interaction Centre, University of Canterbury, Christchurch, New Zealand. ✉e-mail: vanessa.morris@canterbury.ac.nz; christoph.goebl@otago.ac.nz

formation follows a non-seedable primary nucleation-elongation mechanism via a partly-structured dimeric intermediate species. We find that amyloid fibrils of p16 are disaggregated by addition of reducing agent, showing that the disulfide bond is essential for formation and also for the stability of amyloids. While amyloid p16 is unable to inhibit the CDK4 kinase, it regains kinase-inhibiting functionality upon disassembly through reducing agent. When screening multiple cancer-related variants, including mutations remote to the CDK4/6-binding interface, we observe that each mutant shows greatly increased amyloid formation propensity. In contrast, stabilizing mutations were found to prevent amyloid formation, highlighting the importance of amyloid formation in the folding landscape of p16. This study introduces an amyloid-formation mechanism and highlights the

disulfide bond as a specific regulatory element in the formation and stability of an amyloid structure.

## Results

### p16 undergoes rapid amyloid formation upon oxidation

We previously reported that p16 fibrils have a β-sheet, amyloid structure using methods including electron microscopy, diagnostic dyes (Thioflavin-T, Congo Red), Fourier-transform infrared spectroscopy and β-sheet secondary chemical shifts from solid-state NMR spectra[7]. Here, we confirmed the amyloid state of oxidised p16 with X-ray fiber diffraction (Fig. 1b). Pelleted, unaligned fibril samples gave rise to a strong reflection at ~4.6 Å and a weaker more diffuse reflection centered at ~9.9 Å, consistent with the inter-strand and inter-sheet (steric

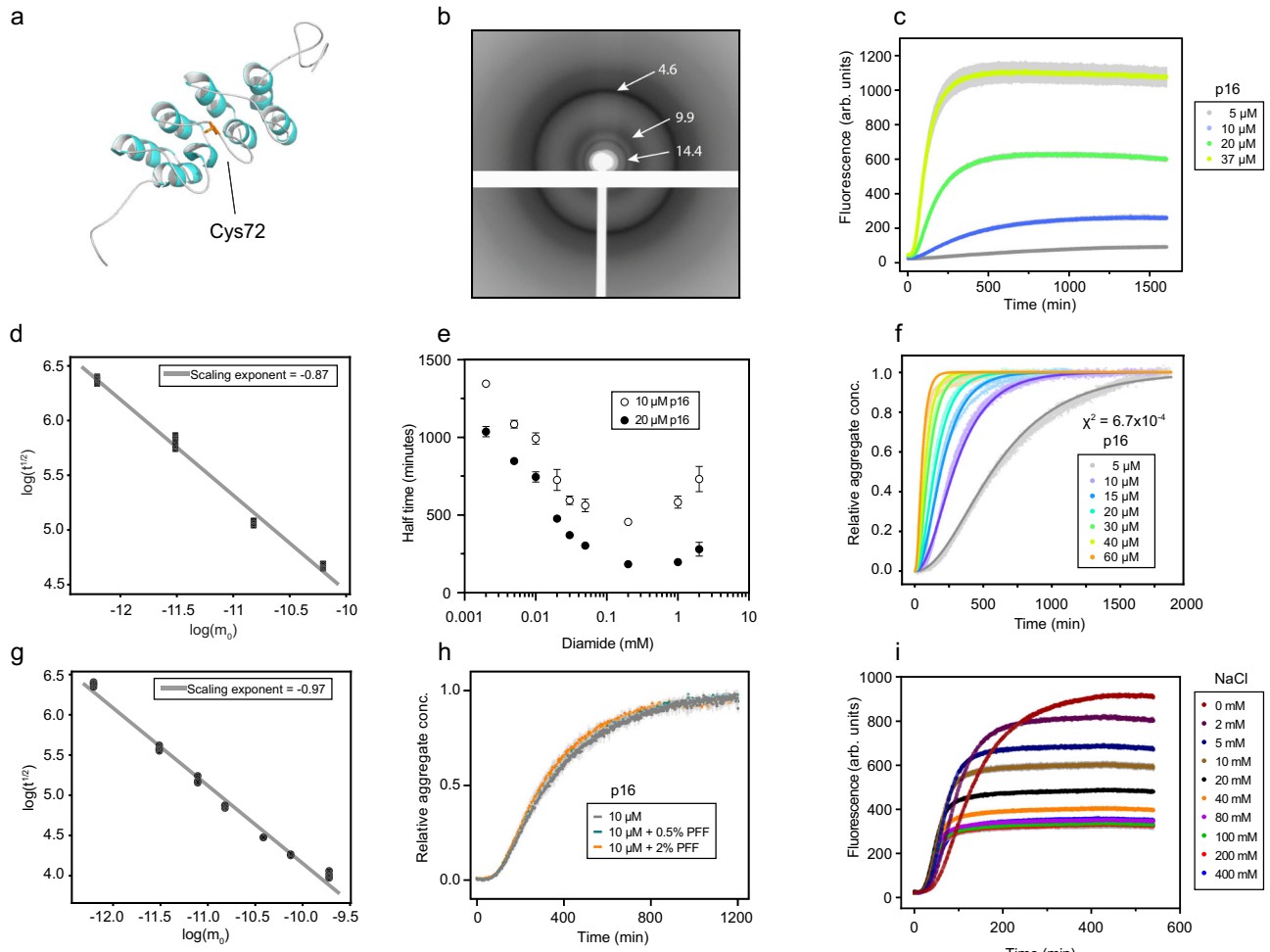

**Fig. 1 | Kinetic analysis of the p16 amyloid formation mechanism. a** Cartoon representation of the p16 structure (PDB accession number 2A5E) highlighting the surface-accessible cysteine residue Cys72[11]. **b** X-ray fiber diffraction image with reflections (labelled in Ångström) produced by p16 fibrils. **c** Aggregation kinetics of p16 as a function of monomer concentration monitored by ThT fluorescence. The series contains protein concentrations from 5−37 μM oxidized with 200 μM diamide. The assay was conducted in 4 mM HEPES pH 7.4 at 25 °C (standard conditions), solid lines represent the mean from four measurements with gray error bars representing standard deviation. **d** Double logarithmic plot of aggregation half times ($t^{1/2}$) (data in Fig. 1c) as a function of p16 monomer concentration ($m_0$) determined by AmyloFit[15]. **e** Half times of amyloid formation of 10 μM or 20 μM p16 oxidized with diamide concentrations ranging from 2 μM – 2 mM. Symbols represent the average of four measurements with error bars representing standard deviation. **f** Aggregation kinetics of p16 as a function of monomer concentration monitored by ThT fluorescence. Aggregation of monomer p16 was initiated at $t_0$ by addition of diamide at a 1:10 molar ratio with respect to protein monomer

concentration. The assay was conducted under standard conditions and symbols represent data from four measurements with fitted solid lines from the nucleated polymerization model (global fitting parameter $k_+k_n = 4.03 \times 10^8$ M$^{-nc}$ h$^{-2}$) of the AmyloFit webserver. **g** Data of Fig. 1f in a double logarithmic plot of aggregation half times ($t^{1/2}$) as a function of p16 monomer concentration ($m_0$) ranging from 5 to 60 μM. Fitting the relationship between these parameters yields a slope of −0.96 and the linearity suggests the dominant mechanism of amyloid formation is unchanged across the monomer concentration range. **h** Seeding does not influence the kinetics of p16 amyloid formation. Pre-formed fibril (PFF) addition of 0.5% or 2% monomer equivalents does not change the lag phase of 10 μM p16 monomer folding into amyloid. Gray error bars represent standard deviation from four measurements. **i** Aggregation kinetics of 20 μM p16 as a function of NaCl concentration monitored by ThT fluorescence. p16 aggregation was initiated by addition of 200 μM diamide in the presence of 0–400 mM NaCl in addition to standard conditions. Gray error bars represent standard deviation from four measurements. Source data are provided as a Source Data file.

zipper) spacings in the cross-β structure characteristic of amyloid. A sharp reflection was also observed at 14.4 Å; the origin of this remains to be determined.

We next characterized the formation of p16 amyloids using Thioflavin-T (ThT) kinetic analysis[14]. For this, we developed optimized protocols for p16 amyloid formation assays to achieve reproducible data. Due to the fast amyloid-formation kinetics, oxidant was added immediately before the start of the measurement. The plate was shaken for 10 s initially to guarantee mixing of the components, and the assay was then performed under quiescent conditions, to avoid artifacts that may be introduced by shaking[15]. We used the cysteine-specific oxidant diamide to induce the formation of an intermolecular disulfide bond to exclude the impact of non-specific oxidation of other amino acids[16]. To confirm that ThT does not impact formation kinetics, the reporter dye was added at different time points, and no differences were observed (Supplementary Fig. 1a). Consistent with our previous findings, no amyloid formation was observed in the absence of the oxidant (Supplementary Fig. 1b), and the same flat baseline was found for the C72S mutant with or without diamide[7] (Supplementary Fig. 1b).

In order to investigate the amyloid formation mechanism, we performed three experiments varying the concentration of different components: first, varying protein concentration with constant oxidant; second, varying the oxidant concentration with constant protein, and third, varying both while keeping a constant diamide-to-protein ratio. In the first experiment, lower protein concentration resulted in longer half-times (Fig. 1c, d) and lower fluorescence intensities. These data were analyzed using the AmyloFit routine to study amyloid formation mechanisms, however, none of the various models showed reasonable fits (Supplementary Fig. 2a, d)[15]. Second, varying the diamide concentration yielded a maximum formation rate at about 1:10 protein-to-oxidant ratio, and these rates declined at lower and at higher ratios (Fig. 1e, Supplementary Fig. 2c). This was the case for both 10 μM and 20 μM protein (Fig. 1e). The lower rate at higher oxidant ratio is likely explained by diamide forming a covalent bond to a thiol and then reacting with a thiolate resulting in a disulfide bond and a final hydrazine product[16]. High concentrations lead to accumulation of the diamide-bound species with less available thiolates to react with. In the third experiment, the protein-to-oxidant ratio was kept constant at 1:10 (Fig. 1f). This changed the shape of the ThT curves and yielded the best fit with the nucleation elongation model of AmyloFit (Fig. 1f, Supplementary Fig. 2d), which involves primary nucleation and elongation processes. The linear dependence between half time of formation and monomer concentration (on a log-log plot) indicates that the dominant mechanism of p16 amyloid formation is unchanged within the range of concentrations tested without saturation or competition factors (Fig. 1g)[15].

Next, we studied the nucleation process of p16 amyloid formation and tested whether pre-formed fibrils act as seeds to increase the aggregation kinetics. Oxidant was removed from the pre-formed amyloid sample by dialysis and fibrils were sonicated prior to addition. Diamide was added again immediately before measurement. Surprisingly, there was no major impact on the formation rate with addition of 0.5% and 2% seeds (Fig. 1h, Supplementary Fig. 2e), suggesting that the addition onto the fibril end is not rate-limiting in this system.

Buffers lacking salt have been reported as the best conditions for monomeric p16 stability[11]. We tested p16 amyloid formation in the presence of increasing concentrations of NaCl, and found that higher ionic strength strongly increased the formation rates up to ~40 mM (Fig. 1i, Supplementary Fig. 2b). The cysteine residue is located in a positively-charged region and the salt effect may be explained by more efficient dimerization due to shielding of charges at the dimerization interface, or due to stabilization of the oligomeric and amyloid structure. This behavior had previously been predicted when electrostatic effects of filamentous protein aggregation processes of insulin and PI3K-SH3 were analyzed. Although dependent on the nature of the ions

and structural contributions, a general trend was observed that increased ionic strength leads to faster elongation rates, but at high salt concentrations these rates decreased by destabilizing the protein[17]; we observe the same trend for p16 amyloid formation.

## Amyloid formation is initiated by an oxidation-triggered mechanism via a dimeric, amyloid-prone intermediate

When oxidizing p16 with diamide for 24 h and analyzing the sample using non-reducing SDS-PAGE, a higher molecular weight band was observed, indicating the formation of an oxidation-based dimer (Supplementary Fig. 3a). This species was converted back into monomers by addition of cysteine reducing agents. No cross-linking was observed for the C72S mutant, confirming the role of cysteine side-chain oxidation. The presence of the disulfide bond was also confirmed by mass spectrometry after trypsin digestion of a 24 h post-oxidation sample (Supplementary Fig. 3b). We next performed a time series of p16 oxidation and monitored the dimerization by non-reducing SDS-PAGE. It should be noted that larger (aggregated) species of p16 dissolve in the SDS loading buffer. To ensure the absence of oxidation during sample processing, we blocked free cysteines with N-ethylmaleimide (NEM). Analysis of the gels revealed that the intermolecular disulfide bond is formed during the first 2 h and the dimeric population subsequently stays constant (Fig. 2a). Under our standard conditions, the half time of dimerization was approximately 60 min with no major changes observed after 2 h. To allow direct comparison of oxidation rates with ThT assays, we also performed these experiments in 96-well plates and found the same behavior as in 1.7 ml microcentrifuge tubes. In the presence of salt, we found that the dimerization occurred at a higher rate (Supplementary Fig. 3c). We further observed that the dimerization was protein-concentration dependent, with slower formation at lower protein concentrations (Supplementary Fig. 3d) as expected for a second-order process, and at lower temperature (Supplementary Fig. 3e).

After establishing the measurement of the monomer-dimer transition, we aimed to analyze the larger species formed upon oxidation. We monitored amyloid formation of p16 by non-reducing native PAGE. After 24 h of oxidation some large species had formed that were incapable of entering the gel, presumably corresponding to amyloid fibrils. In contrast, when the cysteine residue was removed by mutation (C72S), no change in size was observed upon addition of oxidant (Supplementary Fig. 3f). This method was used to monitor the time course of the amyloid formation process. For this, we oxidized samples for varying times and treated them with NEM to prevent further oxidation (Fig. 2d). There was a clear shift in molecular weight within the first 2 h, which most likely represents the monomer to dimer transition, with a similar transition as observed by SDS-PAGE. After 1 h oxidation, intermediate oligomeric species at higher molecular weights in the range of hundreds of kilodaltons were observed and these signals increased over time. After 5 h oxidation, an amyloid band was detected, and this band intensified over time. Generally, a clear shift to higher molecular weight species can be observed over the time period tested.

Size-exclusion chromatography is an alternative method to study the formation of high molecular weight species. We oxidized p16 under standard conditions (room temperature and in the absence of salt), and then analyzed 100 μl of the reaction mixture by chromatography at 4 °C, at which temperature further dimerization and amyloid formation are significantly slowed down. It was possible to separate the monomer and dimer from the larger species under these conditions. This analysis showed similar kinetics and formation of larger species as in the native PAGE analysis (Fig. 2b). Although conversion to amyloid was complete within approximately 8 h as shown by the previous ThT analyses, we observed the presence of monomeric and dimeric species by size-exclusion chromatography even at a time point when the ThT signal had plateaued. Quantification of these individual

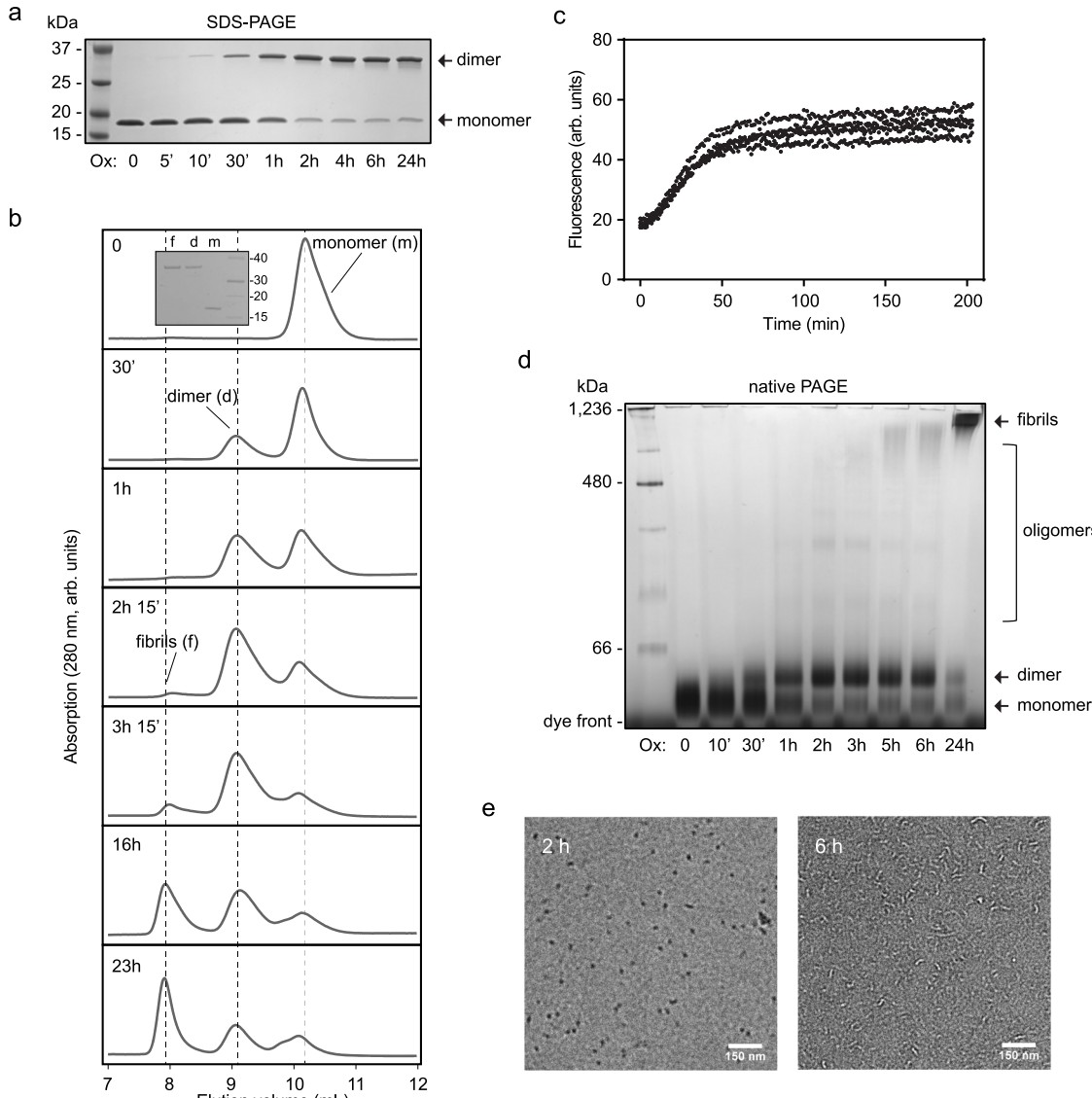

**Fig. 2 | p16 amyloid fibrils form via a dimeric intermediate. a** Non-reducing SDS-PAGE analysis shows gradual accumulation of dimeric p16 triggered by oxidation. The dimeric proportion reaches a maximum by about 2 h following treatment of 20 μM p16 with 200 μM diamide. A single representative of five experiments is shown. **b** Separation of 20 μM p16 reaction mixtures by size-exclusion chromatography after treatment with 200 μM diamide for the indicated duration. The SDS-PAGE inset indicates the monomer/dimer composition of each fraction. A single representative of two experiments is shown. **c** ThT fluorescence analysis starting from dimeric p16 (~5 μM) isolated by size-exclusion chromatography. Four independent measurements are plotted and data are presented without normalization. **d** Native PAGE analysis of a 20 μM p16 reaction mixture after oxidation with 200 μM diamide reveals oligomerization of p16 dimers leading to accumulation of high molecular weight species. A single representative of three experiments is shown. **e** Negative-stained transmission electron micrographs of 20 μM p16 treated with 200 μM diamide after 2 and 6 h oxidation. Representative micrographs are shown from three experiments. Source data are provided as a Source Data file.

species showed the time-dependent conversion of monomer into dimer and larger species, as shown in Supplementary Fig. 4a.

Together, these data suggest that the oxidation-induced dimer is the active species that assembles into amyloid fibrils. We confirmed this by isolating the dimeric species from the reaction mixture using size-exclusion chromatography and subsequent ThT analysis demonstrating amyloid formation. Due to dilution on the column and experimental constraints, only small volumes of dilute samples could be obtained using this method. To obtain larger amounts, we optimized the sample preparation and chromatography conditions, with inclusion of NaCl in the running buffer to improve peak separation. We found that the isolated dimer was stable at 4 °C and did not further evolve into amyloid (Supplementary Fig. 4b). When performing the ThT assay with the dimeric fraction in the absence of oxidizing agents and at room temperature, we indeed observed

formation of amyloid (Fig. 2c) that was much faster than at a similar concentration when starting from monomer and oxidizing agent (compare Fig. 1f).

Aggregation of p16 amyloid fibrils was also monitored using transmission electron microscopy (TEM), where monomers and dimers are too small to be visualized, but fibril structures can be observed due to their large dimensions. In order to directly compare the imaging data to ThT and size-exclusion chromatography assays, size exclusion chromatography fractions containing the amyloids were used to prepare TEM samples. After 2 h of oxidation, only small species with poor contrast were visible, while at 4 h short fibrils were observed. The fibrils appeared to gradually lengthen by 6 and 8 h, with average fibril lengths changing from 34.7 nm at 4 h, to 40.8 nm at 8 h (Fig. 2e, Supplementary Fig. 4c, d). Further morphological changes were seen over longer time points, with longer fibril lengths observed in samples

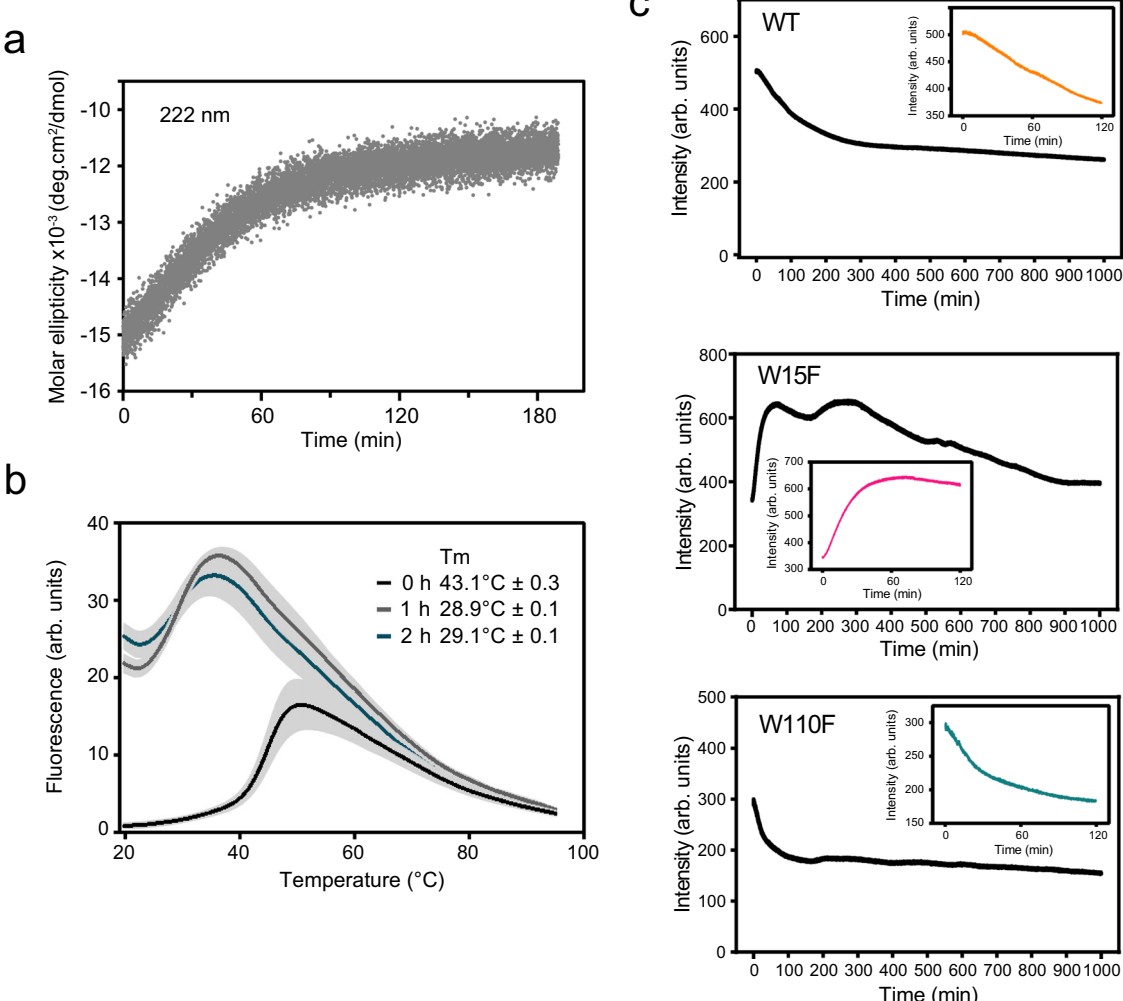

**Fig. 3 | Structural changes accompany dimerization of p16 following oxidant exposure. a** Circular dichroism spectroscopy analysis of p16 following oxidation under standard conditions. The molar ellipticity at 222 nm increases the most during the first hour following oxidation. **b** Differential scanning fluorimetry (DSF) analysis reporting thermal unfolding by binding and fluorescence of a SYPRO Orange dye. Error bars in gray represent standard deviation from the average of five measurements. The melting temperature (Tm) reported including standard deviation was derived from the maximum of the first derivative of the curves. **c** Intrinsic tryptophan fluorescence spectroscopy analysis of wild-type, W15F and W110F p16 following oxidation. The inset shows the initial 120 min of the measurement. A single representative of three experiments is shown. Source data are provided as a Source Data file.

after 72 h of oxidation. These changes in fibril structure, even after time points when the ThT signal plateaus, suggest that there may be an equilibrium of fibrils with smaller species to allow for rearrangement of amyloids. Alternatively, it can also be envisioned that short fibrils could merge through end-to-end interactions. This process is usually considered negligible for typical amyloid systems[18], but its rate depends on the length distribution of the fibrils and it could become significant for very short fibrils over longer periods.

## Characteristics of the transient dimeric species

We next characterized the structural transition into dimers induced by oxidation. The far-UV circular dichroism (CD) spectrum of p16 in phosphate buffer showed the presence of an α-helical protein, as reported earlier[19] (Supplementary Fig. 5a). To maintain standard measurement conditions among all our kinetic data, we performed time-dependent measurements on p16 in HEPES buffer, despite slight signal interference from the buffer at low wavelengths. Upon oxidation for 4 and 24 h, the signal became less intense but retained minima at 208 and 222 nm that are characteristic for α-helical structures, suggesting residual helices present in the intermediate and possibly in the amyloid state (Supplementary Fig. 5b). Resuspending the protein

solution for homogenization immediately before the measurement did not impact the signal, suggesting that the signal was not lost due to sedimentation of larger species. To study the early dimerization process in more detail, we measured the time-dependent CD signal upon oxidation at 222 nm only (Fig. 3a). We detected a steep decrease in intensity for the first hour, indicating a change of structure upon dimerization. We further characterized the initial transition into dimer by acquiring melting curves by differential scanning fluorimetry (DSF) using SYPRO Orange as a reporter dye. A relatively low thermal unfolding temperature of 42 °C has previously been reported for p16 based on CD measurements[19] and in line with this, our measurements of monomeric p16 yielded an unfolding temperature of 43.1 ± 0.3 °C. After the sample was oxidized for 1 h and 2 h, when the majority of the protein is dimeric, clear melting transitions of 28.9 ± 0.1 and 29.1 ± 0.1 °C were detected, indicating that the dimer was less stable than the monomer (Fig. 3b). The presence of oxidant did not alter the melting temperature of the C72S mutant (Supplementary Fig. 5c). DSF measurements of an amyloid sample after 24 h of oxidation showed no transition, but the curve showed high background fluorescence typical of aggregated proteins with exposed hydrophobic residues[20] (Supplementary Fig. 5d).

Together, these data suggest that oxidation generates a partially-structured dimer that subsequently folds into the amyloid state. To gain further insights into this state, we measured the intrinsic tryptophan fluorescence intensity over the initial time course of oxidation. This method has been successfully applied to amyloid formation assays previously[21,22] and the two naturally-occurring tryptophan residues offer site-specific insights. The wild-type protein showed a fast change of the intrinsic fluorescence intensity for the first 120 min, followed by a slower decay (Fig. 3c). We then created two constructs, each with one tryptophan mutated into phenylalanine, W15F and W110F. Both showed highly similar unfolding temperatures compared to the wild-type protein as probed by DSF, indicating that the structures and stability were not disrupted by the point mutations. Detection of fluorescence intensity upon oxidation showed similar time-dependent changes but these were positive for W15F and negative for W110F, indicating different local changes for each residue (Fig. 3c). Increases in fluorescence intensity typically indicates decreased solvent exposure[23]. Measurements of tryptophan emission maxima showed only small changes and are therefore not reported.

## Reversible p16 amyloids rely on a regulatory disulfide bond

Because the p16 amyloid transition is triggered by oxidation and involves disulfide cross-linked species, we investigated whether the dimeric and amyloid species are stable upon addition of reducing agents. First, we induced amyloid formation (monitored by ThT) and added an excess of reducing agent once the amyloid formation appeared to be complete. The ThT signal of the amyloids in the plateau phase started to decrease immediately upon addition of the reducing agent, returning almost to the initial intensity level (Fig. 4a) and suggesting disassembly of the fibrils. When adding the reducing agent during the amyloid formation period, the signal continued increasing for a short period before it reached an early maximum and eventually dropped. We next isolated the dimeric species by size exclusion chromatography and added DTT immediately in the beginning of a ThT assay. Whereas the amyloid starts to disassemble upon addition of DTT (Fig. 4a), the isolated dimer produced a strong increase in ThT signal before decreasing back to the initial plateau (Fig. 4b). The dimer therefore appears, to some extent, resistant to reduction and might only disassemble from the amyloid conformation.

The conversion of the amyloid species into monomers upon addition of reducing agent was also studied through size-exclusion chromatography. Figure 4c shows the reverse process compared to Fig. 2b, in which the large species disassemble over time into monomers with no dimeric species visible. This data was confirmed with electron microscopy analysis in which amyloids are only visible in the oxidized state and are absent after reduction (Fig. 4d).

To investigate whether amyloid p16 recovers its original fold and functionality upon reduction, we measured functional and aggregation assays. The major function of monomeric p16 is inhibition of CDK4. We confirmed the regulation of this kinase by p16 using an activity assay, in which CDK4 was supplemented with ATP and its substrate, the protein retinoblastoma (Rb), and the phosphorylated Rb product was detected with an antibody (Fig. 4e). As expected, the presence of 5-fold unoxidized, monomeric p16 inhibited this reaction, whereas amyloid p16 did not. Nevertheless, the inhibition was restored when reduced, disassembled amyloids were added. The same effect was also observed at a 2-fold p16-to-kinase ratio and controls show that presence of oxidant (diamide) or reductant (DTT) do not interfere with the reaction (Supplementary Fig. 6). We further confirmed the redox-based reversibility of the amyloid state by using size-exclusion chromatography to isolate the reduced monomers. We found that oxidation of the reduced protein leads to reconversion into amyloid with the same rate as a fresh monomeric control (Fig. 4f). When the amyloid p16 species was isolated by size-exclusion chromatography and incubated with reducing agent before 10 mM NEM was added to stop the reaction

and derivatize excess DTT, the SDS-PAGE gel analysis shows a slow disulfide bond reduction over several hours (Fig. 4g). These experiments highlight the full reversibility of the p16 amyloids through disulfide reduction.

## Amyloid formation of p16 by hydrogen peroxide is enhanced by bicarbonate

To broaden our analysis to physiological oxidants, we next focused on hydrogen peroxide, a by-product of the respiratory chain and an important secondary messenger that has been shown to regulate numerous signaling pathways[24,25]. This oxidant undergoes a reversible reaction with bicarbonate, leading to the formation of peroxymonocarbonate, a molecule found to be 1000-fold more reactive towards some proteins[26–29]. We therefore examined $H_2O_2$ and peroxymonocarbonate as potential oxidants for triggering p16 amyloid formation.

A bicarbonate buffer system was freshly prepared and the pH adjusted before addition of hydrogen peroxide. At 50 μM $H_2O_2$, which represented a protein-to-oxidant ratio of 1:2.5 (Fig. 5a), no major transition was observed for the $H_2O_2$ alone, whereas the presence of bicarbonate clearly led to amyloid formation. Since bicarbonate addition introduces ions and p16 amyloid formation is increased in the presence of salt, we performed the same experiment with NaCl in place of the bicarbonate buffer (both with similar ionic strength and 25 mM concentration). The presence of NaCl did not lead to amyloid formation under these conditions (Fig. 5a).

We then varied the oxidant concentration, performing the same experiments using 200 μM (Fig. 5b) and 1 mM $H_2O_2$ (Fig. 5c) (1:10 and 1:50 protein-to-oxidant ratio, respectively). In the presence of the higher hydrogen peroxide concentration, slow amyloid formation was also observed in the absence of the bicarbonate buffer. As before, addition of the bicarbonate buffer greatly increased the amyloid formation rate (~6 times faster) and this effect was also greater than in the control experiment with NaCl (Fig. 5b, c). We confirmed that the bicarbonate buffer alone did not induce amyloid formation in comparison to treatment with 200 μM $H_2O_2$ (Supplementary Fig. 7c). Electron microscopy indicated that amyloids formed in this system had the same curved morphology as those formed by diamide (Fig. 5d).

In order to gain further mechanistic insights, we next performed ThT assays in the presence of hydrogen peroxide and bicarbonate, with varying protein concentrations (10 to 40 μM) while keeping the protein-to-oxidant ratio fixed (1:2.5, 1:10 and 1:100) (Supplementary Fig. 7d). A linear correlation was observed between the logarithmic half-time and the protein concentration. This behavior was also observed for diamide-induced oxidation and suggests that the mechanism does not vary between different protein concentrations (Supplementary Fig. 7a). The data were analyzed with AmyloFit, and we found that at a 1:100 protein-to-oxidant ratio, the data yielded a reasonable fit to the nucleation elongation model, as was the case for oxidation with diamide[15]. In contrast, at lower protein-to-oxidant ratios, no AmyloFit model was able to fit the data (Fig. Supplementary 7d). Given that the models of protein aggregation implemented in AmyloFit do not contain a chemical modification step of the protein, such as the dimerization in our case, it is plausible that our data can only be fitted under conditions in which the kinetics of the dimerization is so rapid that it is no longer rate-limiting.

To study only the initial oxidation step, SDS-PAGE was again used to monitor protein dimerization. A clear difference in dimerization kinetics was observed upon oxidation with hydrogen peroxide in the presence and absence of bicarbonate buffer, suggesting that peroxymonocarbonate is a more reactive oxidant for p16 disulfide bond formation (Fig. 5e). Dimerization was also measured at lower and higher protein concentrations in the presence of bicarbonate and with a 10-fold molar excess of hydrogen peroxide. We observed a

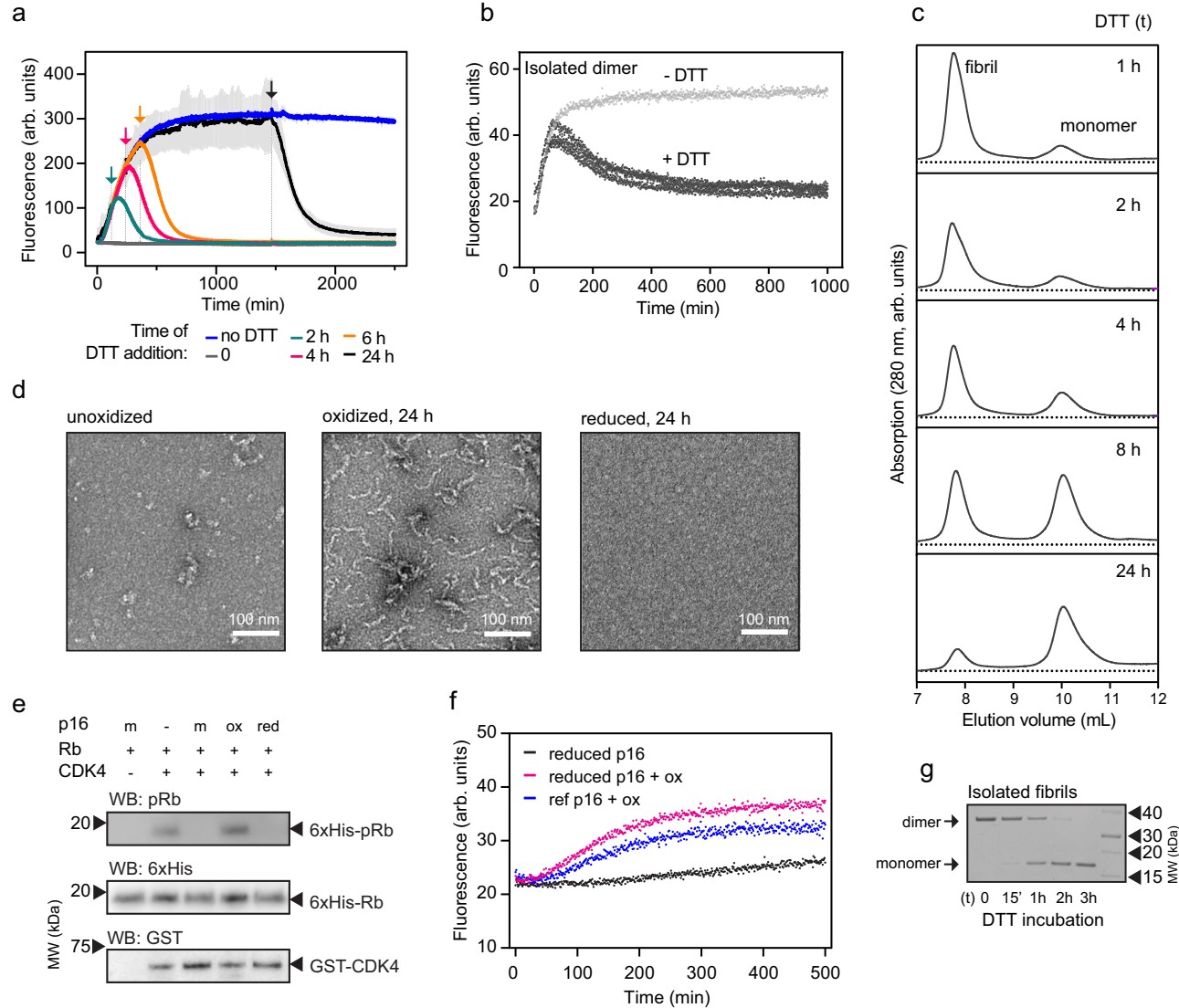

**Fig. 4 | p16 amyloids are disassembled by reduction of disulfides. a** Addition of excess DTT (2 mM) indicated by arrows causes reduction of ThT fluorescence to baseline levels. The amyloid formation is prevented by DTT addition at $t_0$ (dark gray). The assay was conducted under standard conditions using 20 μM p16 and 200 μM diamide. Gray error bars represent standard deviation from the average of four measurements. **b** ThT-monitored amyloid formation of isolated p16 dimer (black signals, three replicates) is impacted by addition of excess DTT at time zero. The data in gray shows amyloid formation without addition of DTT. **c** SEC of p16 species after addition of 2 mM DTT to oxidized p16 for the indicated durations. **d** Negative-stained electron micrographs (L to R) of 20 μM untreated p16, after 24 h oxidation with 200 μM diamide, and oxidized p16 after 24 h treatment with 2 mM DTT. Small, amorphous aggregates are frequently observed for unoxidized p16 samples and are assumed to represent small amounts of protein precipitate

due to unfolding/degradation. Single micrographs are representative of three experiments. **e** Kinase assay of CDK4/Cyclin D1, Rb and ATP in the presence of 5-fold excess of monomeric p16 (m), amyloid p16 (ox) and reduced amyloids (red). Oxidation and reduction were performed for 24 h at room temperature. Representative results are shown from two experiments. **f** ThT analysis of isolated monomer (5 μM) derived from DTT-treated amyloid (pink) after addition of 10-fold diamide, the sample in black is lacking oxidant addition. A reference sample of monomer (no prior treatment with oxidant or reducing agent) treated with oxidant at the same concentrations was measured for comparison (blue). Data are the average of four replicates. **g** SDS-PAGE of amyloids incubated with 2 mM DTT showing the conversion of disulfide-bonded dimer to monomers. Representative gel images are displayed from two experiments. Source data are provided as a Source Data file.

concentration dependence similar to that found with diamide oxidation (Supplementary Fig. 7b). For both oxidants, we observed that the dimerization step was almost at completion at the time that amyloid fibrils were halfway through formation as measured using ThT assays.

### Cancer-related and stabilizing p16 mutants show greatly altered amyloid formation propensity

p16 is amongst the most mutated proteins in cancer and mutations have so far mainly been studied in the context of its binding site to cyclin-dependent kinases (CDK4 and CDK6)[9] (Fig. 6a). These kinases are the major cellular interaction partners of p16 and together they regulate cell division; modification of this complex formation is

considered one of the hallmarks of cancer formation[10]. For this study, we produced ten recombinant protein variants harboring previously-reported single-point p16 cancer mutations that are distributed over the structure. Five mutants (A20P, G35A, D84G, H98Y and P114L) did not yield structured monomeric protein and were therefore not included in the further analysis; the structural integrity of the remaining five mutants was confirmed by DSF (A20S, D84N, V95A, G122S, A127S, Supplementary Fig. 8a) and their amyloid formation was subsequently characterized. We found that four out of five mutants had a lower melting temperature compared to the wild-type (Supplementary Fig. 8a). The half-times of amyloid formation determined by ThT assays are shown in Fig. 6b–d. They were found to

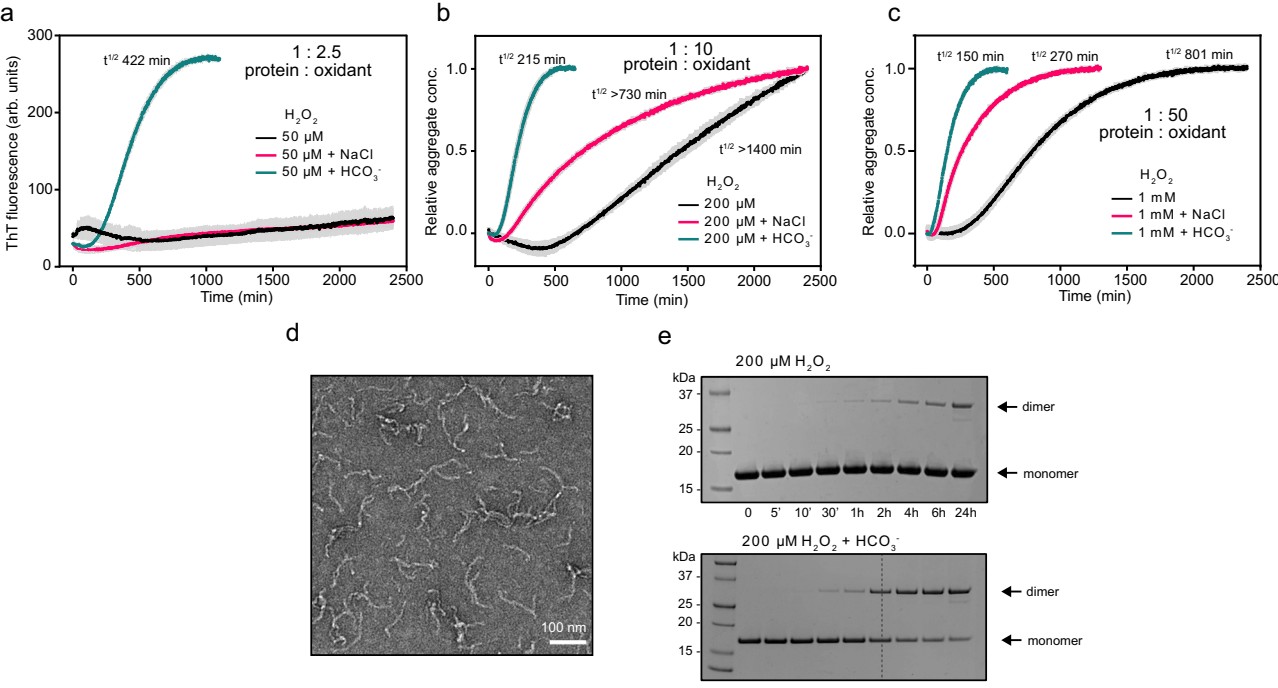

**Fig. 5 | Bicarbonate accelerates p16 amyloid formation by H$_2$O$_2$. a–c** H$_2$O$_2$ pre-incubated in bicarbonate buffer to yield peroxymonocarbonate efficiently triggers p16 amyloid formation. Reactions include 50 µM (**a**), 200 µM (**b**) or 1 mM (**c**) H$_2$O$_2$ in standard p16 buffer (4 mM HEPES buffer pH 7.4) or with added 25 mM NaCl or bicarbonate buffer (25 mM, pH 7.4), and 20 µM p16 protein. The protein to oxidant ratio is shown. Error bars in gray represent standard deviation from the average of four measurements. Data presented in **b** and **c** are normalized. **d** Negative-stained transmission electron micrograph of 20 µM p16 oxidized with 200 µM H$_2$O$_2$ in the presence of 4 mM HEPES, 25 mM bicarbonate buffer for 48 h. Scale bar = 100 nm. A representative image is shown from at least three experiments. **e** p16 dimerization rate measured by non-reducing SDS-PAGE. 20 µM p16 was oxidized with 200 µM H$_2$O$_2$ in the presence (bottom panel) or absence (top) of 25 mM bicarbonate buffer pH 7.4. After the specified treatment time 10 mM NEM was added to remaining monomeric p16 to prevent further dimerization. Representative gel images are shown from three experiments. Source data are provided as a Source Data file.

be between 67 and 81 min, therefore all five cancer variants formed amyloid fibrils with about twice the rate of the wild-type species of 147 mins.

To further characterize the role of protein stability on the amyloid formation mechanism, we produced four additional variants harboring the previously-reported stabilizing mutations W15D, L37S, L121R and a combination of these three mutations, the so-called hyper-stabilized triple mutant (HTM)[30]. These mutations showed slightly increased melting temperatures, consistent with previous reports (Supplementary Fig. 8b). Interestingly, the Thioflavin-T amyloid formation assay did not show any change in fluorescence over the timescale measured for three of the four variants and these mutants therefore did not convert into amyloid (Fig. 6d, e). Only L37S showed a change in ThT intensity over time, but with a half-time of 265 mins, which is nearly twice as slow as the wild-type. These findings were further confirmed by size-exclusion chromatography after oxidation for 24 h, when only very little or no protein was detected in the void volume where wild-type amyloids usually elute (Supplementary Fig. 8c).

We next followed the transition of p16 into amyloid in a cellular environment. For this, we cultured human embryonic kidney (HEK) 293 cells and transiently expressed human wild-type p16, p16 C72S and the cancer mutants A20S, D84N and V95A. Using western blotting, we followed the molecular weight of p16 upon treatment of the cells with hydrogen peroxide, which, in buffer containing bicarbonate, likely leads to formation of peroxymonocarbonate[31]. Wild-type p16 showed a dimeric band upon addition of oxidant that increased over time and incubation with reducing agent resolved the dimer band (Fig. 7a). In order to confirm that oxidized p16 forms larger species in cells, we performed a filter trap assay. Upon gentle lysis of the cells, the

samples were applied to a membrane and any large protein aggregates remained on the surface for antibody detection. An increase in p16 aggregates can be seen in the pelleted fraction after exposure to oxidant and addition of reducing agent leads to disassembly of these large species.

We next tested the dimerization and aggregate formation of the three cancer mutants over short timescales (Fig. 7b). The combined western blot and filter trap analysis show that they have a strong tendency to form large, oxidized species. Interestingly, in the filter trap assay p16 V95A shows the presence of aggregates even without addition of oxidant, suggesting high susceptibility to aggregation. We confirmed that dimers are disulfide-bond linked using a p16 C72S variant that does not show presence of oxidation-dependent aggregates (Supplementary Fig. 9a). Equal protein loading for the filter trap assays was confirmed using Ponceau S staining (Supplementary Fig. 9b). These experiments highlight the impact of cancer-related single-point mutations that show a modified behavior upon exposure to oxidants.

Endogenous oxidant levels vary greatly between different cell types and cancer cell lines are known to generally have elevated oxidant concentrations[32]. We used western blot analysis to search for reducible p16 dimers in different cancer cell lines without the addition of oxidant. We indeed identified a cell line (Ishikawa cells) that shows a band at the size of dimeric p16 (Fig. 7d). Upon addition of the reducing agent β-mercaptoethanol, this band disappears which indicates the presence of oxidized p16, potentially in the amyloid state. Such a band was not identified in a second cancer cell line, HCC-2998 cells (Fig. 7d). Filter trap analysis confirms the presence of large p16 species in Ishikawa cells that can be disassembled by addition of reducing agent (Fig. 7d).

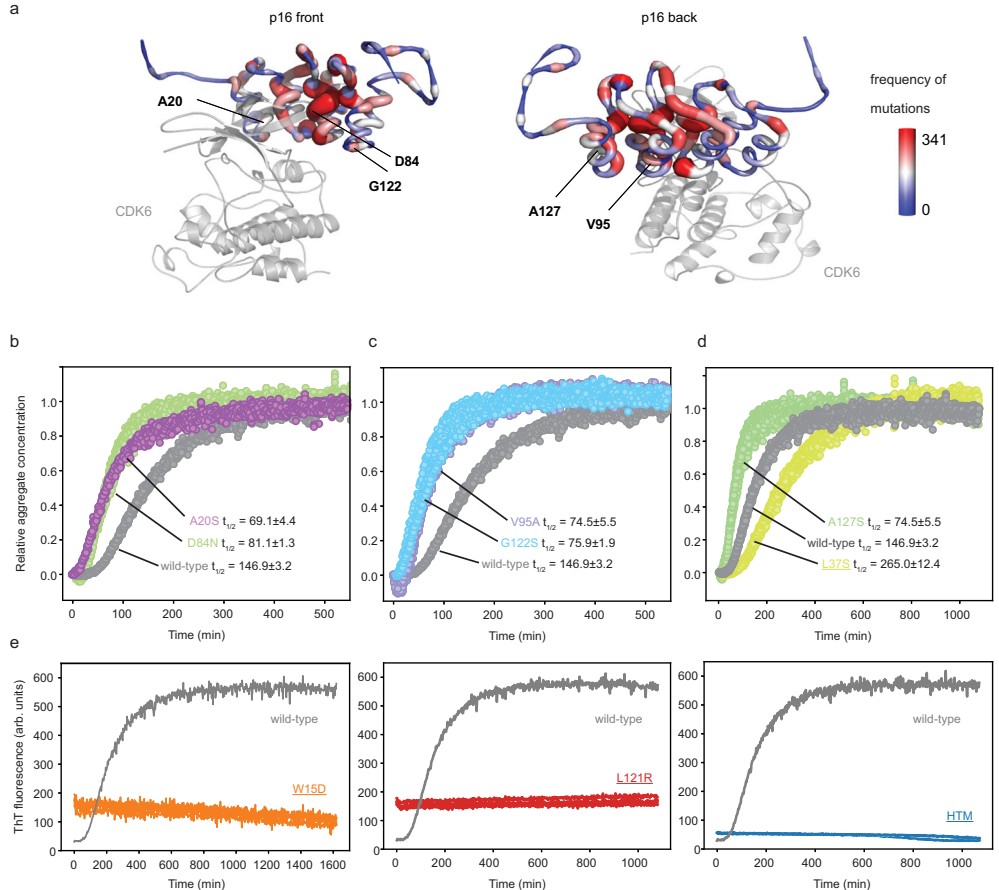

**Fig. 6 | Cancer-associated p16 variants form amyloid at a faster rate than wild-type p16, stabilizing mutations largely prevent formation into amyloid.**
**a** Cartoon representation of the p16-CDK6 complex[9]. CDK6 is shown in gray and semi-transparent. The p16 structure encodes the mutations reported in the COS-MIC database[75] in a sausage plot and color encoded (thin, blue = low number of mutations, minimum = 0, thick, red = high number of mutations, maximum = 341).
**b-d** The ThT analysis of five cancer mutants shows a systematically shorter halftime of amyloid formation, experiments ($n = 4$) were performed in parallel with wild-type using 20 µM protein and 200 µM diamide at standard conditions. **d**, **e** In contrast, the oxidation of individual stabilizing mutations L37S, W15D, L121R (underlined), and their combination in the hyper-stabilized triple mutant (HTM) show much slower or no conversion into amyloid. Source data are provided as a Source Data file.

## Discussion

This study introduces a type of amyloid fibril where both its formation and stability rely on the presence of a regulatory disulfide bond. We have described a sequential mechanism in which oxidation leads to the formation of a structurally unstable, dimeric species that self-assembles into amyloid. These p16 amyloids form quickly and reproducibly at physiological conditions without the need of low pH buffers, high temperature, crowding agents or sample agitation. The protein is a stable monomer prior to oxidation and once formed, amyloids are stable but can be completely disassembled by addition of a reducing agent. The analysis of kinetic curves obtained with ThT fluorescence revealed typical features observed during amyloid formation, including a measurable lag phase and a sigmoidal transition into a stable plateau[33]. A constant protein-to-oxidant ratio yielded half-times of amyloid formation with linear dependence on the protein concentration. These data were best fit by the nucleation elongation model in AmyloFit, suggesting the absence of feedback mechanisms such as secondary nucleation or fragmentation events[15]. Indeed, using our standard setup, seeding of ThT samples with pre-formed fibrils did not strongly impact amyloid formation kinetics. This is a highly unusual behavior for an amyloid system, because the addition of a monomer to a preformed fibril has an almost universally lower activation free energy compared to the formation of a nucleus[34,35]. The fact that seeding effects are found to be negligible here suggests that the free energy barriers for nucleation and elongation are more similar to each

other than in other amyloid systems. Furthermore, the dimerization could be partly rate-limiting and mask the seeding effect. AmyloFit does not explicitly include a kinetic term that takes the formation of the p16 dimer into account. Particularly if the oxidation occurs on the same time scale as the subsequent assembly into amyloid fibrils, the data cannot be expected to be easily fittable with AmyloFit. In agreement with this consideration, we observe that the kinetic data is best fitted for the highest oxidant concentration, where the dimerization step is rapid and ceases to be the sole rate-limiting step.

Here, we used multiple methods to study the initial oxidation event and we characterized a dimeric species that is the key intermediate in amyloid formation. The dimer appears after short incubation periods with oxidant, at a time when the ThT signal remains in its baseline plateau, suggesting that this species does not have a ThT-positive cross-β-sheet structure. This is further supported by the time-dependent CD signal. It shows that the main change during the first hour of oxidation is a slight reduction of the α-helical content, a drop from about 40% to 30%, according to deconvolution using BeStSel[36]. Similarly, intrinsic tryptophan fluorescence showed major changes in intensity during the first hours. When mutating W15F (detecting W110), the fluorescence signal increased, whereas it decreased for W110F (detecting W15) during the first hour. This type of analysis reports the changes in environment for each tryptophan residue; high fluorescence intensities reflect less solvent-based quenching[21,23], whereas decreasing intensities reflect an increase in tryptophan solvent

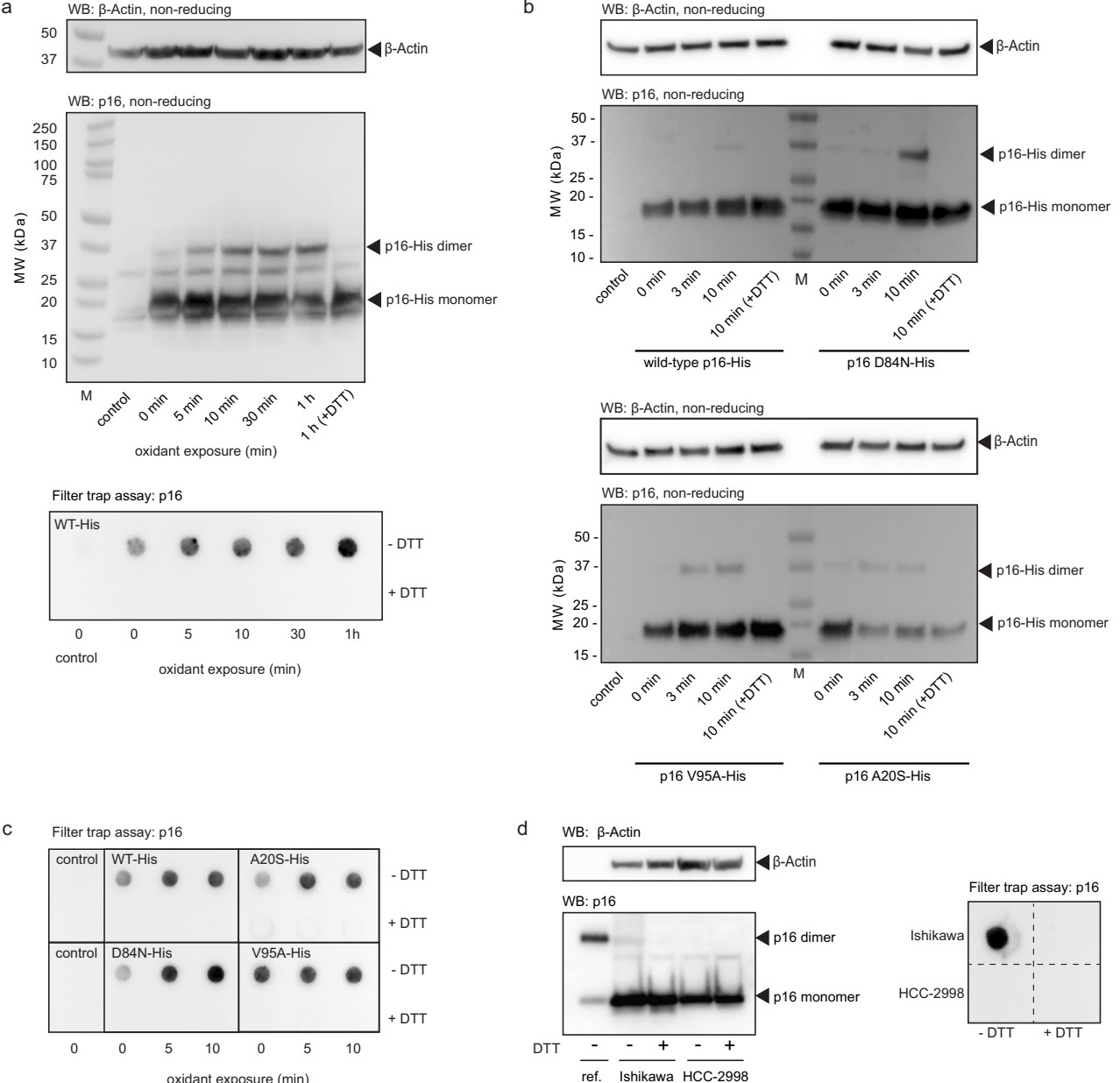

**Fig. 7 | Oxidation of p16 variants in human cell cultures. a** Dimerization and aggregation analysis of transiently expressed wild-type 6xHis-tagged p16 (p16-His) in HEK293 cells by western blotting and filter trap assays. Cells were exposed to 200 µM hydrogen peroxide in media and reactions were stopped at the times indicated by addition of 10 µg/mL catalase and 50 mM NEM. An untransfected control sample (control) was included. The samples labeled (R) and +DTT were incubated with 10 mM DTT. β-actin blotting was included as a protein loading control. This blot is representative of three experiments. **b**, **c** Short-term oxidant exposure of hexa-histidine tagged wild-type p16 and D84N, V95A, and A20S variants. The cells were exposed to hydrogen peroxide for 3 or 10 min, and the same control conditions were included as in (**a**). Samples were analyzed by western blot (**b**) and filter trap assays (**c**). Western blots are representative of two experiments. **d** Lysates from Ishikawa and HCC-2998 cell lines immunoblotted for p16 and β-actin (loading control) in the presence (+) and absence (-) of reducing agent (5% β-mercaptoethanol); a filter trap assay was performed under the same conditions. Representative blot images are displayed from two experiments. An oxidized recombinant p16 standard is included (ref.) for comparison.

accessibility. At one- and two-h post-oxidation, p16 is mostly dimeric, and DSF characterization showed that melting temperatures were reduced by about 12 °C. Together, these data suggest that newly-oxidized p16 dimers are still partly α-helical. It was previously found that p16 unfolding by denaturants follows a sequential pattern with the first N-terminal ankyrin repeat being less stable than the three C-terminal repeats[37]. Our tryptophan fluorescence findings are consistent with this observation, indicating a less stable N-terminal repeat upon dimer formation. W15, which is located in the first repeat, displays decreased fluorescence upon oxidation, suggesting a change of

structure into a more solvent-exposed conformation such as local unfolding. As C72 is located in the ankyrin-type L-shaped long loop region, the oxidized p16 dimer must have its dimer interface located at this region. W110 is situated in the C-terminally neighboring loop and the tryptophan fluorescence data suggests that it forms part of the hydrophobic dimer interface. Combining the site-specific fluorescence data with the loss of α-helical structure and a lower melting temperature, these observations might suggest unfolding of the first ankyrin-repeat region of the protein and formation of a dimer interface along the long loop regions through oxidation. Indeed, an artificial protein

consisting of only three ankyrin repeats has been suggested as the minimum necessary stable repeat-mer for ankyrin repeat proteins[38]. A structured dimeric interface is also supported by our finding of the high stability of the dimer in the presence of the reducing agent. The SDS-PAGE analysis showed that monomer bands are only visible after about 1 h of DTT addition, suggesting that the disulfide bond is largely excluded from the solvent.

Time-dependent analysis of the gel-filtration separated species showed the gradual transition from monomer into dimer and its conversion into amyloid. Application of native PAGE allowed further refinement of the analysis and revealed discrete bands of oligomeric p16 species forming from about 2 h post-oxidation, a time at which oligomers were also observed by electron microscopy. Formation of p16 amyloids can therefore be studied with methods that are often not suitable for other systems.

Biochemical protein amyloid dissociation studies mainly rely on addition of chemical denaturants or chaperones[39–41]. Here, we find that p16 amyloids are not only dependent on a regulatory disulfide bond for their formation, but also for stability. When reducing agent is added to p16 amyloids, we observe a gradual drop in ThT fluorescence intensity that reaches close to baseline. We speculate that the slow decay of the signal is due to a terminal dissociation mechanism, whereby protein units dissociate from fibril ends while the core of the fibril stays intact. SDS-PAGE analysis suggested that the dimer reduction rates are similar to the ThT signal loss, which could support the terminal dissociation hypothesis. In this case, disulfide bonds would not be accessible throughout the amyloid, rather just at the termini. A disulfide reduction and a specific dissociation step would then lead to the disassembly of the amyloid. At this point, it is unclear if the dissociating unit is a dimer or a reduced monomeric species.

When reducing agents are added to the reaction mixture during buildup of the amyloid, we find that the ThT signal continues to increase for some time before reaching an early maximum and slowly dropping to initial levels with a sigmoidal decay of the signal. This behavior is also found by size-exclusion chromatography and the monomeric species fully re-converts into amyloid upon oxidation. Our functional results using CDK4 highlight the full reversibility of p16 amyloid formation in which both structural and functional features are regulated through oxidation and reduction reactions. Disaggregation mechanisms of amyloid structures in cells have mainly been described through involvement of chaperone systems[42], and it would constitute a less costly mechanism to regulate p16 amyloids through redox reactions in a cellular environment.

In biological systems, various reactive oxidative species (ROS) are produced, with hydrogen peroxide being the most widely studied as a thiol oxidant in cell signaling[25,43,44]. We therefore tested whether the physiologically relevant $H_2O_2$ converts p16 into amyloids, both on its own and in bicarbonate buffer, where it is partially converted to peroxymonocarbonate[45]. The oxidation and amyloid formation rate triggered by $H_2O_2$ was greatly increased in the presence of bicarbonate. Peroxymonocarbonate has been shown to be selective in enhancing the reactivity with certain other proteins such as PTPs[27,45], and our results imply that this is also the case with p16. Besides being highly reactive, the peroxymonocarbonate ion includes a negative charge and the positively-charged surroundings of the p16 cysteine residue may partly explain the strongly-increased reactivity of peroxymonocarbonate.

The role of p16 in cancer has been explored since its discovery in the 1990s[8,46,47]. It suppresses the activity of CDK4/6 that regulate cell division and has been found to be mutated in almost half of known cancers[46]. So far, research has focused on mutations in the binding interface to CDK4/6, where cancer-related mutations have been shown to critically impair the cell-division inhibiting interaction[48–50]. Our analysis showed that single-point mutations can have a strong impact on the fold of the protein and five of the ten cancer mutations tested

did not yield structured protein. Although melting temperatures were only weakly decreased, the remaining correctly-folded variants showed greatly increased amyloid formation rates, indicating a lower energetic barrier to amyloid conversion. Whereas the D84N mutant is known to inhibit CDK4/6 binding, the remaining four mutations are located remotely to the CDK4/6 binding site and their ability to inhibit the kinases was found to be indistinguishable from wild-type p16 in in vitro assays[49]. The role of these mutations in cancer is currently not understood. However, our results show that these variants have greatly increased amyloid formation rates, suggesting a higher tendency to misfold. Since amyloid p16 loses its kinase inhibition capacity, these mutations serve as an example for the potential implication of the p16 amyloid formation mechanism[7]. Our findings are supported by cellular experiments showing that mutant p16 readily transitions into dimeric and large aggregate species. p16 amyloid formation shows some similarity to the aggregation of the tumor suppressor p53, where mutations also lead to a higher tendency to aggregate, though oxidation is not reported to play a role[51–54]. The results for the p16 cancer mutations raise the question if p16 mutations generally increase amyloid formation propensity by destabilizing the evolved structure. To test this, we investigated several stabilizing mutations. In stark contrast to the cancer mutations, we found a complete loss of amyloid formation upon oxidation for three of the four variants. This highlights how single amino acid substitutions have a strong impact on p16 stability and suggests a potential evolutionary role for the instability of p16.

The cysteine residue of p16 is largely conserved within many vertebrate species[7], which could suggest a functional conversion into amyloid. Several features of p16 amyloid formation are consistent with growing observations in the field of the characteristics of functional versus pathogenic amyloid. p16 forms amyloid fibrils rapidly, within minutes to hours, as opposed to most pathogenic amyloid proteins which require hours to days to convert. The mechanism of nucleation-elongation dominated formation is also more commonly observed for functional amyloids, where conversion often needs to take place quickly and without formation of multiple intermediates, while pathogenic amyloids such as amyloid beta typically involve secondary nucleation which may generate toxic side-species[55–57]. It has also been suggested that disulfide bond formation in amyloids may stabilize fibrils and intermediates in such a way that reduces toxicity of the aggregates, as has been shown for lysozyme[58].

Disulfide bonds are known to play critical roles in amyloid formation[59]. In particular, the loss of stabilizing disulfide bonds in monomeric structures is frequently associated with conversion into amyloids, and accelerated fibril formation is observed upon reduction of disulfides for example in lysozyme[60], SOD1[61], insulin[62,63], and the peptide hormone somatostatin-14[64,65]. A strict dependence on intermolecular disulfide bonds for amyloid formation has so-far not been described, however, in some cases amyloid formation is accelerated by their presence[59,66–71]. One newly-described amyloid protein that has some parallels in its formation mechanism to p16 is the fish protein, β-parvalbumin[66]. The aggregation of this protein also follows a primary nucleation/elongation-dominated mechanism, and the amyloid formation can also proceed via a disulfide-bonded dimeric intermediate. Removal of the cysteine residue of β-parvalbumin slows but does not prevent aggregation[72]. Golgi-associated plant pathogenesis related protein 1 (GAPR-1) is a mammalian protein that can form amyloids, and cysteine residues are suggested to regulate activity of the protein. GAPR-1 aggregates in the presence of heparin, and this is enhanced by the presence of copper, which catalyzes disulfide bond formation[73]. Recently, intramolecular disulfide bond formation of an isoform of the Alzheimer's disease-related protein Tau was shown to lead to formation of fibrils that cannot cross-seed fibrils formed by reduced forms of proteins[67]. The disulfide-bonded fibrils could be partially disassembled by addition of reducing agents. The results of the study suggested that

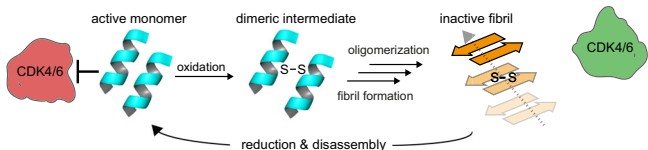

**Fig. 8 | Model of the oxidation-induced amyloid formation and disassembly mechanism of p16.** Helices in blue represent the native fold of p16 and the orange represents refolded fibrillar β-sheets with the stabilizing intermolecular disulfide bond. In the amyloid state, p16 is unable to inhibit CDK4/6 (green). Reduction of disulfide bonds triggers amyloid disassembly and refolding to the active p16 monomer which is capable of inhibiting CDK4/6 (red).

the intramolecular disulfide bond changes monomer structure or dynamics in a way that leads to an alternative fibril formation pathway.

p16 adds to a growing number of amyloid-forming proteins whose structure and function is "switched" by an intermolecular disulfide bond. We note, however, that p16 is the only known amyloid in which intermolecular disulfide bond formation is essential for conversion to fibrils, with no detectable fibril formation in the absence of oxidants, and with complete disaggregation and functional reversibility after reduction of the disulfide bond (Fig. 8). The redox status therefore strictly controls the structural state and function, making p16 a fully redox-switchable system.

## Methods

### Protein expression and purification
The wild-type and mutant protein was expressed as a fusion with an N-terminal hexa-histidine leader sequence and a protein A tag followed by a tobacco etch-virus (TEV) protease cleavage site. Upon cleavage, a glycine-alanine sequence remains N-terminal of the starting methionine residue and amino acid numbering reported here refers to the original protein sequence (Uniprot ID P42771). *E. coli* BL21(DE3) cells harboring the kanamycin-resistant pET-vector were grown at room temperature to an $OD_{600}$ of 0.5 and protein production was then induced using 0.5 mM IPTG at 19 °C for 18 h. Cells were harvested by centrifugation and pellets were resuspended in purification buffer (20 mM HEPES, 110 mM sodium acetate, 5% (v/v) glycerol, 2 mM ß-mercaptoethanol (BME), pH 8.0) and lysed by sonication. After centrifugation, the lysate was then purified by Nickel-NTA resin (Qiagen) following the manufacturer's instructions and using purification buffer including 20 mM imidazole. The protein was eluted by purification buffer containing 220 mM imidazole and then applied to size-exclusion chromatography (ÄKTA Pure, HiLoad 16/600 Superdex 75 pg column, Cytiva) using purification buffer. Protein fractions containing tagged p16 were pooled and digested overnight at 4 °C by addition of -1:50 tobacco etch virus protease. The sample was then applied to Nickel-NTA resins and the flow-through fraction contained the purified p16 protein. After desalting into p16 buffer (4 mM HEPES, pH 7.4) using a HiPrep 26/10 Desalting column (Cytiva), protein aliquots were concentrated to ~40 μM and stored at −80 °C. All constructs were obtained from Genscript and purified using the same method.

### Amyloid preparation
For ThT and PAGE analysis, fibrils were generated by quiescent incubation of monomeric p16 with diamide or $H_2O_2$ at room temperature in 4 mM HEPES, pH 7.4. For reactions containing sodium bicarbonate, 125 mM buffer (pH 7.4) was prepared immediately prior to a 10 min preincubation with $H_2O_2$ to generate peroxymonocarbonate. The oxidant/buffer solution was then added to p16 at a ratio of 1:4 to give a final sodium bicarbonate concentration of 25 mM. Salt increases the rate of p16 amyloid formation, so to control for the salt effect we performed the same procedure but instead with NaCl at the same concentrations as above. For seeding experiments, fibrils were produced by treating 20 μM p16 with 200 μM diamide for 24 h at room temperature. To remove the oxidant, sample was transferred to a Slide-A-Lyzer dialysis device (Thermo Scientific), 3.5 K MWCO, and dialyzed against 4 mM HEPES buffer pH 7.4 at 4 °C overnight on a stir plate with intermittent buffer exchanges. The sample was then recovered and sonicated at 20% amplitude for 20 s (0.5 s on, 0.5 s off) on ice using a UP200S Ultrasonic Processor (Hielscher). The concentration of seed samples was measured using a nanodrop 1000 spectrophotometer before dilution and addition to ThT seeding experiments.

For kinase assays, amyloid p16 was prepared by incubating purified recombinant monomer protein (30 μM) with a 10-fold excess of diamide for 24 h at room temperature in 500 μL total volume. The oxidized sample was then applied to an ÄKTA pure chromatography system with a Superdex 200 10/300 GL column (Cytiva) at 4 °C running with 4 mM HEPES, pH 7.4. A flow-rate of 0.6 mL/min was applied and protein species were detected using the absorbance at 280 nm. Fractions containing p16 amyloid were collected and protein concentration was determined using a NanoDrop 8000 Spectrophotometer (Thermo Scientific) before removing a partial volume for reduction with 2 mM dithiothreitol (DTT) overnight. Monomeric p16, SEC-isolated amyloid p16 and reduced SEC-isolated amyloid p16 proteins were then evaluated for their ability to impede CDK4/cyclin D1 activity in an in vitro kinase assay. Samples for SDS-PAGE dissociation experiments were performed the same way with SEC-purified amyloids and incubation with 2 mM DTT for 15 mins, 1 h, 2 h and 3 h followed by addition of 10 mM *N*-ethylmaleimide (NEM) for 10 mins before addition of SDS sample buffer and running of the gel.

### X-ray fiber diffraction
Fibrils of p16 in 4 mM HEPES, pH 7.4 were produced by treatment of 30 μM monomeric protein with 300 μM diamide for 72 h at room temperature. Protein fibrils were desalted into water using a 5 mL HiTrap 26/10 Desalting column connected to an ÄKTA pure chromatography system (Cytiva), and concentrated using a 10 kDa MWCO Ultra Centrifugal filter (Amicon). Droplets containing p16 fibrils were suspended between the wax-filled ends of two glass capillaries and allowed to air-dry, resulting in short fibril stalks. X-ray fiber diffraction images were obtained using a Rigaku XtaLAB Synergy Custom system with FR-X Cu source and HyPix-6000HE hybrid pixel array detector. Fibers were optically aligned and then still X-ray diffraction images were collected with CrysAlisPro, which was also used for image generation.

### SDS-PAGE and native PAGE
After treatment with oxidant, p16 samples were blocked with 10 mM NEM to prevent further dimerization before species were resolved by PAGE. For SDS-PAGE analysis samples were combined with denaturing sample buffer with or without reducing agent (DTT or BME) and resolved on a 4–15% gradient Mini-Protean TGX Stain-Free (Bio-Rad) or Bolt 4–12% Bis-Tris Plus (Invitrogen) gel at 200 V. Molecular weights were estimated by comparison to Precision Plus Protein Dual Color Standards (Bio-Rad) or Novex Sharp Pre-stained Protein Standard (Invitrogen). The native PAGE method was adapted from Schägger and von Jagow[74]. NuPAGE Bis-Tris 4–12% gradient polyacrylamide gels (Invitrogen) were pre-run with 0.5x TBE buffer before loading protein samples containing 10% glycerol. Gels were run at a constant current of 25 mA for 60 min using the Native-PAGE Running Buffer with Cathode Buffer Additive (Invitrogen). Molecular weights of protein bands were estimated by comparison to NativeMark Unstained Protein Standard (Invitrogen).

### Circular dichroism spectroscopy
Far-UV CD spectra were measured on a Jasco J-815 CD spectrometer with a data pitch of 0.2 nm, a scanning speed of 50 nm/min, and a digital integration time of 1 s at room temperature. A reference spectrum was obtained of 5 μM p16 in phosphate buffer while scans after

oxidation were taken of 20 µM p16 in HEPES buffer. A 1 mm pathlength quartz cuvette was used for the measurements at room temperature. Five scans were taken per sample and buffer signal was subtracted for background correction. Measurements of kinetic analysis of loss of α-helical signal following p16 oxidation was performed by recording at 222 nm for 180 min. Data units were converted from millidegrees (m°) to molar ellipticity (deg.cm²/dmol) using a mean residual weight of 105.45 g/mol.

## Differential scanning fluorimetry

Thermal stability of p16 after oxidation was monitored by DSF. Five replicates per condition were measured in a 96-well 0.2 mL MicroAmp optical 96-well reaction plate (Invitrogen) containing 30 µL of 20 µM p16 and 5x SYPRO Orange (Invitrogen). Fluorescence was recorded on a QuantStudio 3 real-time PCR machine during a 0.015 °C/s heating ramp from 20–95 °C. Data was processed and analyzed using Protein Thermal Shift Software 1.4. Melting temperature was obtained for each replicate by evaluation of the maximum of the first derivative of acquired data. The time delay between oxidant addition and beginning the temperature ramp was estimated to be 3 min.

## Intrinsic tryptophan fluorescence

p16 wild-type or tryptophan mutants (20 µM) were oxidized with 200 µM diamide in 4 mM HEPES pH 7.4. Tryptophan fluorescence was measured in a Cary Eclipse fluorescence spectrophotometer (Agilent) in a 0.5 mL quartz cuvette. The excitation wavelength was 280 nm and emission was measured at 355 nm, with measurements taken every 1 s. Slit widths were set to 2.5 nm (excitation) and 5 nm (emission) for wild-type p16 and 5 nm for both excitation and emission for W15F and W110F p16.

## Analytical and quantitative size-exclusion chromatography

To assess the time-dependent oligomeric state of p16 following oxidation a series of samples were subject to size-exclusion chromatography. For analytical measurements, 100 µL of 20 µM p16 samples oxidized with 200 µM diamide and incubated at room temperature were applied to an ÄKTA pure chromatography system with a Superdex 200 10/300 GL column (Cytiva) at 4 °C running with 4 mM HEPES, pH 7.4. Unicorn 7 (Cytiva) software was used for data collection. The same conditions were used to analyze the time-dependent decrease of amyloid species following treatment with DTT. For this, p16 was oxidized under standard conditions for 24 h and 2 mM DTT was added to promote amyloid disassembly. Samples were loaded using a 1 mL disposable syringe and applied using a 500 µL or 1 mL loop. A flow-rate of 0.6 mL/min was applied and protein species were measured using the absorbance at 280 nm. For identification of the oxidation state, samples of interest were collected and applied to SDS PAGE.

For isolation of the dimeric species, 1 mL of 20 µM protein was oxidized with 200 µM diamide for 1.5 h at room temperature using 4 mM HEPES, pH 7.4 buffer. The sample was then applied to size exclusion performed at 4 °C including 40 mM NaCl in the sample buffer to improve peak separation, yielding about 250 µL of a ~5 µM dimer sample. SEC purified amyloid species were prepared the same way but after incubation for 24 h. For isolation of monomeric p16 derived from DTT-treated amyloid, SEC purified amyloids were incubated with 2 mM DTT overnight at room temperature. The sample was again applied to size exclusion and the resulting monomer fractions were collected for ThT analysis. An untreated sample of p16 from the same initial stock solution was diluted in SEC running buffer to the same concentration for reference. Elution profiles monitoring absorbance at 280 nm were compared against a monomeric control.

## Thioflavin-T fluorescence assays

In standard conditions, 50 µL samples consisting of 20 µM p16 in 4 mM HEPES, pH 7.4 were measured in the presence of 10 µM ThT and oxidant was added immediately before the start of the assay. Four replicates for each condition were measured in a 96-well half area plate (Corning) covered with a MicroAmp Optical Adhesive Film (Thermo Fisher Scientific) at 25 °C. Samples were subjected to one initial 10 s mixing period before obtaining measurements (Ex/Em 435/482 nm) at 1–2 min intervals. Measurements were made on a Molecular Devices M5 microplate reader in bottom read mode and data were collected using SoftMax Pro software. Data were normalized for kinetic and half-time analysis using the AmyloFit webserver[15].

## Negative-stain transmission electron microscopy

Carbon-coated copper grids (ProSciTech) were floated on a 6 µL droplet of 20–50 µM p16 in 4 mM HEPES, pH 7.4 for 60 s. Grids were washed once with water and placed on a 7 µL droplet of 2% (w/v) uranyl acetate stain for 30 s. Excess stain was blotted with filter paper and grids were dried overnight before measurement on a Philips CM-200 transmission electron microscope. Fibril lengths were measured manually using ImageJ, with 120 particles measured over three micrographs for each time point.

## HPLC/Mass spectrometry

Tryptic digestion of an amyloid p16 sample was performed at a 50:1 substrate:trypsin (Promega) weight ratio for 12 h at 37 °C using a 20 µM protein sample. Peptides were analyzed by LC-MS/MS using a Thermo Scientific Velos Pro ion trap mass spectrometer coupled to a Dionex UltiMate 3000 HPLC system with a 50 µL injection loop. The sample was stored on the autosampler tray at 5 °C prior to analysis. A Jupiter 4-µm Proteo 90 Å column (150 × 2 mm, Phenomenex, Torrance, CA) at 40 °C was used for chromatographic separation using water and 0.1% formic acid as Solvent A and acetonitrile and 0.1% formic acid as Solvent B. The column was equilibrated with 95% Solvent A and 5% Solvent B for 5 min then a linear gradient was run for 40 min to 50% Solvent A and 50% Solvent B to achieve separation. 35 µg of digested p16 amyloid was injected and the flow rate was set at 0.2 mL/min. Nitrogen was used as the sheath gas and the temperature of the heated capillary was 275 °C. Identification and characterization of peptides was performed using a double play method, first obtaining the full mass spectrum ($m/z$ 300–1000) in positive-ion mode followed by fragmentation of the most abundant ions with helium gas collision-induced dissociation. Data analysis was performed with Thermo Xcalibur Qual Browser 2.2 SP1.48 (Thermo Fisher Scientific). Peptide fragments were manually assigned based on Roepstorff-Fohlman nomenclature.

## Cell culture and western blotting

Ishikawa cells were cultured in minimum essential medium (MEM)-alpha medium (Gibco) supplemented with 5% fetal bovine serum (FBS), HCC-2998 cells were cultured in RPMI 1640 medium (Gibco) supplemented with 10% FBS, and HEK293 cells were cultured in Dulbecco's Modified Eagle Medium (DMEM) medium (Gibco) supplemented with 10% FBS. All media contained 100 µg/mL penicillin and 100 units/mL streptomycin. Cells were maintained in a humidified incubator at 37 °C and 5%done $CO_2$/air.

Cells were harvested at 80% confluency by dissociation with trypsin (TrypLE Express, Gibco) for 5 min at 37 °C or until cells had fully detached from the flask. Trypsin was neutralized with fresh media and cells were collected and centrifuged at 1,000 g for 5 min. Pelleted cells were lysed immediately in buffer containing 62.5 mM Tris-HCl pH 6.8, 2% SDS, 10% glycerol and 125 units/mL benzonase nuclease (Sigma, E1014). After centrifugation at 15,000 g for 10 min to remove insoluble material, the protein concentration of cell lysates was estimated using the *DC* Protein Assay kit (Bio-Rad) and 125 µg total protein was combined with denaturing loading buffer and boiled for 5 min. Reducing samples contained 5% BME. Samples were resolved on a 4−12% Bolt™ Bis-Tris Mini Protein Gel (Invitrogen) run at 200 V for 30 min. Proteins

were transferred to a polyvinylidene fluoride (PVDF) membrane for 60 min at 100 V using transfer buffer containing 20% methanol. The membrane was blocked in 5% (w/v) non-fat milk in tris-buffered saline with Tween® 20 (TBST) buffer for 60 min, then incubated with anti-p16 monoclonal antibody (ab108349) at 1:1000 dilution in blocking buffer overnight at 4 °C. After thoroughly washing the membrane with TBST, horse-radish peroxidase (HRP)-conjugated goat anti-rabbit secondary antibody (Dako, Agilent) at 1:10,000 was applied for 60 min. After further washes, detection of membrane-bound antibody was performed using the Amersham ECL Select Western blotting reagents (GE Healthcare) and visualized using the UVItec Q9 Advanced Chemidoc. Source data are provided as a Source Data file.

For experiments involving the over-expression of p16 in HEK293 cells, the primary antibodies used were anti-p16 monoclonal antibody (Abcam ab108349) 1:1000 and anti-β-actin (Sigma A5310) 1:80,000. HRP-conjugated secondary antibodies used were goat anti-mouse (Dako P0447) 1:2500, goat anti-rabbit (Dako P0448) 1:10,000.

For the kinase assay, the primary antibodies used were anti-p16 monoclonal antibody (Abcam ab108349) 1:1000, anti-phospho-Rb (Ser807/811) (Cell Signaling #9308) 1:1000, anti-his-tag (Abcam ab18184) 1:1000, anti-GST-tag (Abcam ab36415) 1:1000. Where multiple antibodies were used to probe the same membrane, the membrane was first stripped in buffer containing 100 mM β-mercaptoethanol, 62 mM Tris-HCl pH 6.7, 2% SDS with gentle agitation for 30 min at 50 °C. Membranes were washed thoroughly in TBST before blocking and reprobing.

### Over-expression of p16 in HEK293 cells and oxidant treatment

p16 genes (human ORF Clone CDKN2A) were obtained from GenScript. The genes encoded for wild-type protein and single-point mutated variants C72S, A20S, D84N and V95A containing a C-terminal hexahistidine tag (6xHis) and were cloned into a pcDNA3.1 vector harboring ampicillin resistance. Human plasmid DNA was purified using the Plasmid DNA purification Kit Midi (Macherey-Nagel) following manufacturer's instructions. DNA concentration and purity were measured using a NanoDrop® 1000 (Thermo Fisher) and stored at −80 °C. Transfections were performed using a Neon™ Transfection system 100 μL kit. Cells were washed with phosphate-buffered saline (PBS) buffer and centrifuged twice at 2000 g for 5 min and resuspended in Neon™ buffer R to a cell density of $5 \times 10^6$/mL. For each electroporation procedure, 100 μL cell suspension containing $5 \times 10^6$ cells was added to a sterile Eppendorf tube containing 5 μg plasmid DNA. A Neon® pipette fitted with a 100 μL tip was used for electroporation for 20 ms for each of 2 pulses at 1100 V. Electroporated cells were subsequently expelled into 1 mL pre-warmed antibiotic-free media in 12-well plates. Cells electroporated in the absence of DNA were included as a negative control.

For treatments with oxidant, on the following day, media were exchanged with fresh growth media and allowed to equilibrate for 1 h. Oxidant treatment was performed by the addition of freshly-prepared $H_2O_2$ in media and added immediately to cells for the times stated. Reactions were terminated by the addition of 10 μg/mL catalase and 50 mM NEM in PBS and cells were collected by scraping and centrifugation at 1000 g for 5 min.

### Filter trap assay

After harvest, cell pellets were washed in PBS containing 10 μg/mL catalase and 50 mM NEM and centrifuged before addition of 100 μL of lysis buffer (50 mM Tris at pH 7.5, 1% v/v Triton X-100, 1.5 mM MgCl₂, 5 mM EDTA, 100 mM NaCl, protease inhibitors [Pierce, EDTA-free A32955]) for 10 min at room temperature. The solution was pelleted by centrifugation for 15 min at 15 000 x g and 4 °C. The pellet was resuspended in 100 μL benzonase buffer (50 mM Tris-HCl pH 8.0, 1 mM MgCl₂) and 25 U of benzonase was added followed by incubation

for 1 h at 37 °C. The reaction was stopped by adding 2x termination buffer (40 mM EDTA, 0.2% SDS, with or without 10 mM DTT). Protein content was measured using a *DC* Protein Assay kit (Bio-Rad) and 100 μg of protein per sample was loaded on a Bio-Dot microfiltration apparatus (Bio-Rad) with a 0.2 μm pore-size nitrocellulose membrane (Bio-Rad) soaked in buffer B (10 mM Tris-HCl pH 8.0, 150 mM NaCl, 2% SDS) on top of two Whatmann paper filters soaked in Buffer A (10 mM Tris-HCl pH 8.0, 150 mM NaCl, 1% SDS), after pre-washing the membrane with 100 μL of buffer A. Samples were filtered through the membrane by applying a vacuum, followed by washing twice with 100 μl of buffer A. The nitrocellulose membrane was further processed as for western blotting to detect trapped p16.

### Kinase assay

Kinase activity of CDK4/cyclin D1 in the absence or presence of different structural forms of p16 was assessed by detection of phosphorylated retinoblastoma protein in reaction mixes following incubation of Rb1 protein, ATP, ± p16 and CDK4/cyclin D1. Reaction mixes were prepared in 30 μL volumes and contained 1 μg recombinant hexahistidine-tagged Rb1 (MyBioscience, MBS143254) and 1 mM ATP in kinase buffer (25 mM Tris pH 7.5, 10 mM MgCl₂) ±70 or 175 nM p16 (monomer or SEC-isolated amyloid ± DTT as reducing agent, as described in the amyloid preparation section). The reaction was initiated by the addition of 35 nM CDK4/cyclin D1 (ThermoFisher, PV4400) and incubated for 30 min at 30 °C. The assay was terminated by the addition of reducing sample buffer and samples were boiled for 5 min before loading on a Bolt™ 4–12%, Bis-Tris Mini Protein Gel (Invitrogen) followed by western blotting.

### Reporting summary

Further information on research design is available in the Nature Portfolio Reporting Summary linked to this article.

## Data availability

All data are available in the main text, supplementary materials, or as source data files. PDB structure files referred to are available: 2A5E, 1BI7. Mass spectrometry data has been deposited to FigShare https://doi.org/10.6084/m9.figshare.25920136 [https://doi.org/10.6084/m9.figshare.25920136]. Source data has been deposited to FigShare https://doi.org/10.6084/m9.figshare.25970758 [https://doi.org/10.6084/m9.figshare.25970758]. Source data are provided with this paper.

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

## Acknowledgements

We would like to thank Shaun Mucalo for technical assistance with electron microscopy measurements, Nina Dickerhof for technical assistance with mass spectrometry and Christine Winterbourn for suggesting the hydrogen peroxide and bicarbonate experiments. We thank William Lewis for assistance with X-ray fiber diffraction data collection, carried out at Sydney Analytical, a core research facility at the University of Sydney. We thank Tahlia Whiting, Elsie Duncan and Jorja Quinn for their assistance with preliminary data generation. We thank Christine Winterbourn, Mark Hampton, Nico Tjandra and Alexander Peskin for helpful discussions. We would also like to thank our funding bodies for the following grants: Health Research Council of New Zealand Sir Charles Hercus Health Research Fellowship # 20/137 (CG), Royal Society of New Zealand Marsden Fund project grant # 21-UOO-128 (CG), Canterbury Medical Research Foundation project grant # MPG2021-Morris (VKM) and # MPG2022-Goebl (CG)

## Author contributions

Designed research: S.H., M.S., A.B., V.M., C.G. Performed research: SH, S.G., E.H., K.O., S.B., A.D.B., P.C., N.M., J.N., D.G., A.S., M.S., V.M., C.G. Analyzed data: S.H., S.G., E.H., K.O., S.B., A.D.B., M.S., A.B., V.M., C.G. Writing: S.H., M.S., A.B., V.M., C.G.

## Competing interests

The authors declare no competing interests.
