## [Peer Review File · Nature Communications]

Reviewers' Comments:

Reviewer #1:

Remarks to the Author:

The manuscript by Heath et al. presents a groundbreaking study that elucidates a redox-regulated amyloid system. Within this system, the monomeric p16^{INK4a} (p16) tumor suppressor protein undergoes an amyloid transition upon dimerization via a disulfide bond. Intriguingly, the stability of the formed amyloid seems to hinge on this singular disulfide bond, as evidenced by its reversible disassembly into a monomeric state upon the introduction of a reducing agent. The authors highlight the novelty of their findings, emphasizing that this is the inaugural description of fibril formation where the intricate transition into amyloids, along with their structural stability, is dictated solely by redox reactions and a solitary regulatory disulfide bond.

The findings presented in the paper are indeed riveting, paving new avenues of understanding in the domain. They contribute significantly to the existing body of knowledge. Yet, there are areas that warrant further clarification and examination by the authors:

1. The research could be considerably enriched by incorporating findings on the reversible amyloid disassembly within a cellular milieu. This would provide insights into its physiological or pathological implications. An approach could be the transfection of cellular models followed by observations of amyloid aggregation.
2. The manuscript delineates that numerous cancer-associated mutations heighten the amyloid formation propensity of the p16 tumor suppressor. In light of this, it would be prudent for the authors to reference and engage with previously published studies that have explored other tumor suppressor and cancer-associated proteins (e.g.: *Biochemistry*. 2020 Jan 21;59(2):146-155. doi: 10.1021/acs.biochem.9b00796; *J Biol Chem*. 2012 Aug 10;287(33):28152-62. doi: 10.1074/jbc.M112.340638; *Chem Rev*. 2023 Jul 26;123(14):9094-9138. doi: 10.1021/acs.chemrev.3c00131; *Curr Opin Chem Biol*. 2023 Feb;72:102230. doi: 10.1016/j.cbpa.2022.102230.)
3. There are also previous studies showing the disulfide bond regulating the amyloidogenic processes, such as: *J Biol Chem*. 2014 Jun 13;289(24):16884-903. doi: 10.1074/jbc.M114.548354; *Science*. 2009 Jul 17;325(5938):328-32. doi: 10.1126/science.1173155; and for a review: *Arch Biochem Biophys*. 2022 Feb 15;716:109113. doi: 10.1016/j.abb.2021.109113.

Reviewer #2:

Remarks to the Author:

In the manuscript by Heath et al, entitled "Amyloid formation and depolymerization of the tumor suppressor protein p16^{INK4a} is strictly controlled by an oxidative thiol-based mechanism", the authors carefully describe the amyloid conversion of p16, a protein that is often related to tumorigenesis. In this very particular mechanism, the protein appears to form amyloid fibrils with an intermediary entity which is a dimer, formed by the reversible oxidation of a single cysteine residue present in the protein. When oxidized with diamide, the protein forms dimers that temporally evolve to amyloid fibrils. When reduced with DTT, amyloids are drawn back to monomers. The biological function of the monomers submitted to an oxidation-reduction cycle is probed in an Rb phosphorylation system, coupled to CDK4. Another aspect of the work describes the specific effects of H₂O₂ and a enhanced by bicarbonate combination on p16 oxidation and the effects detected in specific parts of the protein through the site-directed mutation of the tryptophan residues present in its sequence, with local change in solvent exposure and thus different fluorescence intensities, giving hints on the structural changes suffered by the protein after dimerization and amyloid formation. The effect of mutations was also explored and the faster

oxidation and aggregation of the mutant proteins, with a consequent inactivation, explain the deleterious effects observed in cancer.

This paper is an important work in the area, with very clear results in well-performed experiments describing a new oxidation-related protein aggregation that is fully reversible. The text is clear, but sometimes lacks formality, with minor flaws listed below, but one major question remains after round 1 of revision by other referees. The issues that need to be addressed are listed below:

Major

1. The biological relevance of this phenomenon is a very important point to be included in this work. Figure 5f presents WB results that are non-conclusive, since bands are very weak. Could the authors explain the choice of these two cell lines for this experiment? An alternative assay should be presented to clarify this point. For instance, authors should perform the transfection of cells with p16 and one or more of the mutants found in cancer. Then, treatment with a ROS-inducing agent, such as cisplatin, in comparison with a mock condition, and a reducing agent approach such as the one performed here would bring important answers.

Minor

1. For more organized exposure of data, this reviewer suggests the authors to switch the order in panels shown in Figure 3, since figures 3b and c are described first in the results section, and then figure 3a. I understand the thread followed, so it would be easier to change the panels order instead of the text.

2. The use of "where" instead of "in which" and "when" throughout the text is in excess. Please review this for a better comprehension. (Eg. "These findings were further confirmed by size-exclusion chromatography after oxidation for 24 h, where only very little or no protein ...").

3. Correct the sentence on page 8 - "We confirmed the regulation this kinase by p16 using an activity assay...".

4. Figure 6 "d and e, in contrast, the oxidation of stabilizing mutations (underlined) shows much slower or no conversion into amyloid". - What is underlined in panel d? Should L37 be?

5. Authors should include information about HTM in Figure 6 legend.

6. Figure 7 – For a conclusion figure, a better explanation of the ideas presented in the paper should be presented to the readers.

7. Page 14 - The sentence "The discovered mechanism of nucleation-elongation dominated formation is also more commonly observed for functional amyloids, where conversion often needs to take place quickly and without formation of multiple intermediates, while pathogenic amyloids such as amyloid beta typically involve secondary nucleation, which may generate toxic side-species⁵⁰" needs at least a second reference that describes what is inferred to the functional amyloids (Functional amyloids vs. pathogenic amyloids).

Reviewer #3:

Remarks to the Author:

The manuscript by Heath et al., describes investigation of an assembling protein system that is regulated by oxidation of cysteines to form a disulphide bonded dimer which then self assembles. The authors have provided extensive and thorough investigation into the assembly and disassembly under varied conditions of oxidation and reduction clearly showing a redox effect which appears to be fully reversible yielding functional protein following disassembly under reducing conditions. The experimental work includes the use of a physiological oxidant to demonstrate the possibility of this effect being physiological and also show that tumour related variants are more assembly prone. The authors show that a dimer appears in a particular cancer cell line suggesting that this formation is possible in cancer cells.

I have a few comments and questions relating to the work

1) The protein in question is a tumour suppressor protein which begs the question as to whether

the assembly is plays a function or dysfunctional/pathogenic role in cancer.

The authors showed the dimer formation in a cancer cell line but this was a short and rather brief section of the manuscript. Is it possible to demonstrate any assembly in the cells?

2) Importantly, in my review above, I have not mentioned amyloid. Clearly this protein assembles and disassembles. It forms fibrils via an unusual, apparently non-seed dependent mechanism. It forms non-straight filaments. It displays some alpha-helical CD spectra characteristics. At no point in this manuscript do the authors demonstrate a beta sheet formation or demonstrate that this protein forms cross-beta structured amyloid fibrils. This in itself is interesting. The ThT fluorescence alone is not sufficient to show beta-sheet amyloid formation. Many previous papers have shown that ThT fluoresces in the presence of fibrils which do not need to be beta sheet including cross-alpha fibrils and also in viscous solutions. It is really important that this paper includes some further structural analysis showing that the fibrils are amyloid in the manner that amyloid is characterised as cross-beta. If these filaments are alpha-helical it will be very interesting and may explain the reversibility of these fibrils.

A Reference required for AmyloFIT

Point by point response to reviewers

Reviewer comments:

Reviewer #1

The manuscript by Heath et al. presents a groundbreaking study that elucidates a redox-regulated amyloid system. Within this system, the monomeric p16^{ink4a} (p16) tumor suppressor protein undergoes an amyloid transition upon dimerization via a disulfide bond. Intriguingly, the stability of the formed amyloid seems to hinge on this singular disulfide bond, as evidenced by its reversible disassembly into a monomeric state upon the introduction of a reducing agent. The authors highlight the novelty of their findings, emphasizing that this is the inaugural description of fibril formation where the intricate transition into amyloids, along with their structural stability, is dictated solely by redox reactions and a solitary regulatory disulfide bond.

The findings presented in the paper are indeed riveting, paving new avenues of understanding in the domain. They contribute significantly to the existing body of knowledge. Yet, there are areas that warrant further clarification and examination by the authors:

1. The research could be considerably enriched by incorporating findings on the reversible amyloid disassembly within a cellular milieu. This would provide insights into its physiological or pathological implications. An approach could be the transfection of cellular models followed by observations of amyloid aggregation.

Response: We thank the reviewer for their very positive comments. We have now included cellular experiments using transfected p16. For this, HEK293 cells expressing WT human p16 were treated with oxidant and the dimerization state and aggregation properties of p16 were measured using western blots and filter trap assays. The aggregation has been shown to be reversible by the addition of reducing agent to cell lysates. This is now included in the manuscript as Figure 7A and Supplementary Figure 9 and described in the updated manuscript under the results section titled “Cancer-related and stabilizing p16 mutants show greatly altered amyloid formation propensity”.

2. The manuscript delineates that numerous cancer-associated mutations heighten the amyloid formation propensity of the p16 tumor suppressor. In light of this, it would be prudent for the authors to reference and engage with previously published studies that have explored other tumor suppressor and cancer-associated proteins (e.g.: Biochemistry. 2020 Jan 21;59(2):146-155. doi: 10.1021/acs.biochem.9b00796; J Biol Chem. 2012 Aug 10;287(33):28152-62. doi: 10.1074/jbc.M112.340638; Chem Rev. 2023 Jul 26;123(14):9094-9138. doi: 10.1021/acs.chemrev.3c00131; Curr Opin Chem Biol. 2023 Feb;72:102230. doi: 10.1016/j.cbpa.2022.102230.)

3. There are also previous studies showing the disulfide bond regulating the amyloidogenic processes, such as: J Biol Chem. 2014 Jun 13;289(24):16884-903. doi: 10.1074/jbc.M114.548354; Science. 2009 Jul 17;325(5938):328-32. doi: 10.1126/science.1173155; and for a review: Arch Biochem Biophys. 2022 Feb 15;716:109113. doi: 10.1016/j.abb.2021.109113.

Response to 2 and 3: We thank the reviewer for highlighting these studies and we have included and discussed the p53 references in the discussion section titled "Cancer mutations strongly impact amyloid formation propensity" and the disulfide bond references in the discussion section titled "Relevance of oxidation-induced amyloid formation", where we compare p16 amyloids to other similar systems.

Reviewer #2

In the manuscript by Heath et al, entitled "Amyloid formation and depolymerization of the tumor suppressor protein p16INK4a is strictly controlled by an oxidative thiol-based mechanism", the authors carefully describe the amyloid conversion of p16, a protein that is often related to tumorigenesis. In this very particular mechanism, the protein appears to form amyloid fibrils with an intermediary entity which is a dimer, formed by the reversible oxidation of a single cysteine residue present in the protein. When oxidized with diamide, the protein forms dimers that temporally evolve to amyloid fibrils. When reduced with DTT, amyloids are drawn back to monomers. The biological function of the monomers submitted to an oxidation-reduction cycle is probed in an Rb phosphorylation system, coupled to CDK4. Another aspect of the work describes the specific effects of H₂O₂ and enhanced by bicarbonate combination on p16 oxidation and the effects detected in specific parts of the protein through the site-directed mutation of the tryptophan residues present in its sequence, with local change in solvent exposure and thus different fluorescence intensities, giving hints on the structural changes suffered by the protein after dimerization and amyloid formation. The effect of mutations was also explored and the faster oxidation and aggregation of the mutant proteins, with a consequent inactivation, explain the deleterious effects observed in cancer.

This paper is an important work in the area, with very clear results in well-performed experiments describing a new oxidation-related protein aggregation that is fully reversible. The text is clear, but sometimes lacks formality, with minor flaws listed below, but one major question remains after round 1 of revision by other referees. The issues that need to be addressed are listed below:

Major

1. The biological relevance of this phenomenon is a very important point to be included in this work. Figure 5f presents WB results that are non-conclusive, since bands are very weak. Could the authors explain the choice of these two cell lines for this experiment? An alternative assay should be presented

to clarify this point. For instance, authors should perform the transfection of cells with p16 and one or more of the mutants found in cancer. Then, treatment with a ROS-inducing agent, such as cisplatin, in comparison with a mock condition, and a reducing agent approach such as the one performed here would bring important answers.

Response: We have expanded our analysis of p16 cancer mutations by transfection of HEK293 cells with WT p16 and three cancer variants (A20S, D84N, V95A). We followed the p16 state at different time points upon oxidant exposure. The dimerisation and aggregation was analyzed by western blot and filter trap analysis, and these results are shown in the new Figure 7a-c and Supplementary Figure 9a-c. The mutants indeed were found to have a strong tendency to aggregate upon addition of oxidant, and these aggregates were disassembled upon addition of reducing agent.

The Ishikawa cell line shows the presence of endogenous dimeric p16, whereas HCC-2998 is a reference with only monomeric protein. We have expanded on the finding of dimeric p16 in the Ishikawa cell line by performing additional filter trap experiments. The Ishikawa cells showed presence of aggregated p16 that was disassembled upon addition of a reducing agent (new Figure 7d and Supplementary Figure 9d). No such aggregates were present in the HCC-2998 cells. One possible reason why the dimer band was weak in the western blot, is that the aggregates in cancer cells may be less easily solubilised in SDS detergents than are p16 amyloid fibrils formed freshly by addition of oxidant. The aggregates that do not run into the western blot gel could be captured in a filter trap experiment, which is consistent with our new results.

These new experiments are described in the updated manuscript under the results section titled “Cancer-related and stabilizing p16 mutants show greatly altered amyloid formation propensity”. The endogenous p16 analysis has been moved from Figure 5 to the new Figure 7 in a combined cell culture analysis figure.

We think that these experiments greatly expand the scope of our work and we thank the reviewer for their suggestions.

Minor

1. For more organized exposure of data, this reviewer suggests the authors to switch the order in panels shown in Figure 3, since figures 3b and c are described first in the results section, and then figure 3a. I understand the thread followed, so it would be easier to change the panels order instead of the text.

Response: We thank the reviewer for their suggestion. We have switched the order of the panels in Figure 3.

2. The use of "where" instead of "in which" and "when" throughout the text is in excess. Please review this for a better comprehension. (Eg. “These findings were further confirmed by size-exclusion chromatography after oxidation for 24 h, where only very little or no protein ...”).

Response: We have removed several instances of “where” in the text.

3. Correct the sentence on page 8 - "We confirmed the regulation this kinase by p16 using an activity assay...".

Response: This has been corrected.

4. Figure 6 “d and e, in contrast, the oxidation of stabilizing mutations (underlined) shows much slower or no conversion into amyloid”. - What is underlined in panel d? Should L37 be?

Response: We have corrected the figure to show L37S underlined.

5. Authors should include information about HTM in Figure 6 legend.

Response: Information about the HTM mutant has been added to the figure legend.

6. Figure 7 – For a conclusion figure, a better explanation of the ideas presented in the paper should be presented to the readers.

Response: We have created a new conclusion figure, which incorporates the disassembly findings and we have updated the discussion.

7. Page 14 - The sentence "The discovered mechanism of nucleation-elongation dominated formation is also more commonly observed for functional amyloids, where conversion often needs to take place quickly and without formation of multiple intermediates, while pathogenic amyloids such as amyloid beta typically involve secondary nucleation, which may generate toxic side-species⁵⁰" needs at least a second reference that describes what is inferred to the functional amyloids (Functional amyloids vs. pathogenic amyloids).

Response: We have added two references that show a nucleation-elongation dominated mechanism for formation of functional amyloids from the laboratories of Prof. Tuomas Knowles and Prof. Daniel Otzen, which make similar comparisons between mechanisms of functional vs pathological amyloid formation.

Reviewer #3

The manuscript by Heath et al., describes investigation of an assembling protein system that is regulated by oxidation of cysteines to form a disulphide bonded dimer which then self assembles. The authors have provided extensive and thorough investigation into the assembly and disassembly under varied

conditions of oxidation and reduction clearly showing a redox effect which appears to be fully reversible yielding functional protein following disassembly under reducing conditions. The experimental work includes the use of a physiological oxidant to demonstrate the possibility of this effect being physiological and also show that tumour related variants are more assembly prone. The authors show that a dimer appears in a particular cancer cell line suggesting that this formation is possible in cancer cells.

I have a few comments and questions relating to the work

1) The protein in question is a tumour suppressor protein which begs the question as to whether the assembly it plays a function or dysfunctional/pathogenic role in cancer.

The authors showed the dimer formation in a cancer cell line but this was a short and rather brief section of the manuscript. Is it possible to demonstrate any assembly in the cells?

Response: We thank the reviewer for their suggestion. We have now included cellular data where we overexpressed the wild-type protein, the cysteine mutant protein and the cancer mutations A20S, D84N and V95A in HEK293 cells. We show the assembly of this protein into dimers and higher molecular weight species through exogenous addition of hydrogen peroxide for various time points, and this is included in a new figure (Figure 7a-c and Supplementary Figure 9a-c).

We further expanded our analysis of the Ishikawa cancer cell line using filter trap experiments which identified the presence of large p16 assemblies that are reducible upon addition of reducing agent (Figure 7d and Supplementary Figure 9c). These new experiments are described in the updated manuscript under the results section titled “Cancer-related and stabilizing p16 mutants show greatly altered amyloid formation propensity”. The endogenous p16 analysis has been moved from Figure 5 to the new Figure 7 in a combined cell culture analysis figure.

2) Importantly, in my review above, I have not mentioned amyloid. Clearly this protein assembles and disassembles. It forms fibrils via an unusual, apparently non-seed dependent mechanism. It forms non-straight filaments. It displays some alpha-helical CD spectra characteristics. At no point in this manuscript do the authors demonstrate a beta sheet formation or demonstrate that this protein forms cross-beta structured amyloid fibrils. This in itself is interesting. The ThT fluorescence alone is not sufficient to show beta-sheet amyloid formation. Many previous papers have shown that ThT fluoresces in the presence of fibrils which do not need to be beta sheet including cross-alpha fibrils and also in viscous solutions. It is really important that this paper includes some further structural analysis showing that the fibrils are amyloid in the manner that amyloid is characterised as cross-beta. If these filaments are alpha-helical it will be very interesting and may explain the reversibility of these fibrils.

Response: We strongly agree with the reviewer’s point that amyloid fibrils have a specific cross-beta sheet structure, and that it is not possible to confirm formation of amyloid fibrils based on ThT-fluorescence alone. In a previous publication (reference 7), we have shown that p16 assemblies are

indeed beta-sheet amyloid using transmission electron microscopy, Congo Red binding, ThT-fluorescence and Fourier-transform infrared spectroscopy. Importantly, we applied solid-state NMR spectroscopy to the fibril conformation and found that visible core residues were present in the beta-sheet state.

Supplementary Fig 9C of reference 7: ^{13}C Secondary chemical shifts of uniquely assigned amino acid types from a ^{13}C - ^{13}C proton-driven spin diffusion (PDS) magic-angle spinning solid-state NMR spectrum of amyloid p16, showing the presence of β -sheet secondary structural elements. Horizontal lines indicate average values for secondary chemical shifts of alpha-helix (blue) and β -sheet (yellow), and regions are shaded to one standard deviation.

A Reference required for AmyloFIT

Response: We thank the reviewer for their suggestion and we have now included the Amylofit reference.

Reviewers' Comments:

Reviewer #1:

Remarks to the Author:

The revised manuscript by Heath et al. presents a groundbreaking study that elucidates a redox-regulated amyloid system. Within this system, the monomeric p16ink4a (p16) tumor suppressor protein undergoes an amyloid transition upon dimerization via a disulfide bond. Intriguingly, the stability of the formed amyloid seems to hinge on this singular disulfide bond, as evidenced by its reversible disassembly into a monomeric state upon the introduction of a reducing agent. The authors highlight the novelty of their findings, emphasizing that this is the inaugural description of fibril formation where the intricate transition into amyloids, along with their structural stability, is dictated solely by redox reactions and a solitary regulatory disulfide bond.

The findings presented in the paper are novel and pave new avenues of understanding in the field. The authors correctly revised the manuscript according to the comments and criticisms of the reviewers.

Reviewer #2:

Remarks to the Author:

The concerns and suggestions provided by this reviewer were carefully addressed and implemented. Consequently, the paper is now suitable for acceptance.

Reviewer #3:

Remarks to the Author:

I was disappointed that the authors were not able to provide any further data to support the view that the fibrils formed are amyloid. Instead they referred to a previous paper which I don't think supports their assumption that these fibrils are cross beta. This previous paper, whilst extensive, provides data showing fibrils which are curvilinear and do not resemble amyloid fibrils (which are almost always straight). They suggest that the EM, ThT and Congo red support their assumption these are amyloid but as they have agreed, none of these methods provide the level of structural information needed to show something is cross-beta. The ssNMR provides beta and random coil signals. The CD appears alpha-helical. I am afraid that these may not be amyloid fibrils. The FTIR is not conclusive

As I previously said, these fibrillar structures remain really interesting but until the authors can show these are cross-beta, it is not helpful for the field to call them cross-beta amyloid. I would support a discussion of this issue but I am reluctant to support publication in the current form with the title and abstract stating this as an unambiguous conclusion.

REVIEWER COMMENTS

Reviewer #1 (Remarks to the Author):

The revised manuscript by Heath et al. presents a groundbreaking study that elucidates a redox-regulated amyloid system. Within this system, the monomeric p16^{ink4a} (p16) tumor suppressor protein undergoes an amyloid transition upon dimerization via a disulfide bond. Intriguingly, the stability of the formed amyloid seems to hinge on this singular disulfide bond, as evidenced by its reversible disassembly into a monomeric state upon the introduction of a reducing agent. The authors highlight the novelty of their findings, emphasizing that this is the inaugural description of fibril formation where the intricate transition into amyloids, along with their structural stability, is dictated solely by redox reactions and a solitary regulatory disulfide bond.

The findings presented in the paper are novel and pave new avenues of understanding in the field. The authors correctly revised the manuscript according to the comments and criticisms of the reviewers.

We thank the reviewer for their comments.

Reviewer #2 (Remarks to the Author):

The concerns and suggestions provided by this reviewer were carefully addressed and implemented. Consequently, the paper is now suitable for acceptance.

We thank the reviewer for their comments.

Reviewer #3 (Remarks to the Author):

I was disappointed that the authors were not able to provide any further data to support the view that the fibrils formed are amyloid. Instead they referred to a previous paper which I don't think supports their assumption that these fibrils are cross beta. This previous paper, whilst extensive, provides data showing fibrils which are curvilinear and do not resemble amyloid fibrils (which are almost always straight). They suggest that the EM, ThT and Congo red support their assumption these are amyloid but as they have agreed, none of these methods provide the level of structural information needed to show something is cross-beta. The ssNMR provides beta and random coil signals.

The CD appears alpha-helical. I am afraid that these may not be amyloid fibrils. The FTIR is not conclusive. As I previously said, these fibrillar structures remain really interesting but until the authors can show these are crossbeta, it is not helpful for the field to call them cross-beta amyloid. I would support a discussion of this issue but I am reluctant to support publication in the current form with the title and abstract stating this as an unambiguous conclusion.

To address the reviewer's concern, we reached out to Prof. Margaret Sunde from the University of Sydney, who is a well-known expert in amyloid research, for further experimental evidence of p16 amyloids. We acquired X-ray fibre diffraction data of oxidized p16 that show clear reflections at 4.6 and 9.9 Å, consistent with inter-strand and inter-sheet (steric zipper) spacings in the cross-β structure, characteristic of amyloid. This new data has been incorporated in a revised version of the manuscript in Figure 1.

Our previous circular dichroism spectra of fibrils show reduced α-helical content. We assume the residual signal represents α-helical propensity in non-core or flexible regions, noting that α-helical structures show higher intensities than β-sheets per amino acid residue. Regarding the morphology of the fibrils, we note that other curvilinear amyloid fibrils have been reported (see references 1-4 below for examples).

We think that the combined evidence of our previously published results and the newly acquired X-ray diffraction data provide strong evidence of the beta-sheet nature of p16 fibrils. We thank the reviewer for their comments and helpful suggestions to improve this manuscript.

1. Wong YQ, Binger KJ, Howlett GJ, Griffin MD. Methionine oxidation induces amyloid fibril formation by full-length apolipoprotein A-I. *Proc Natl Acad Sci USA* 2010;107:1977–82.
2. Groenning M, Campos RI, Hirschberg D, Hammarström P, Vestergaard B. Considerably Unfolded Transthyretin Monomers Precede and Exchange with Dynamically Structured Amyloid Protofibrils. *Sci Rep.* 2015 Jun 25;5:11443.
3. Wilson LM, Mok YF, Binger KJ, Griffin MD, Mertens HD, Lin F, Wade JD, Gooley PR, Howlett GJ. A structural core within apolipoprotein C-II amyloid fibrils identified using hydrogen exchange and proteolysis. *J Mol Biol.* 2007 Mar 9;366(5):1639-51.
4. Garvey M, Ecroyd H, Ray NJ, Gerrard JA, Carver JA. Functional Amyloid Protection in the Eye Lens: Retention of α -Crystallin Molecular Chaperone Activity after Modification into Amyloid Fibrils. *Biomolecules.* 2017, 7(3):67

Reviewers' Comments:

Reviewer #1:

Remarks to the Author:

The revised manuscript by Heath et al. presents a groundbreaking study that elucidates a redox-regulated amyloid system. Within this system, the monomeric p16ink4a (p16) tumor suppressor protein undergoes an amyloid transition upon dimerization via a disulfide bond. Intriguingly, the stability of the formed amyloid seems to hinge on this singular disulfide bond, as evidenced by its reversible disassembly into a monomeric state upon the introduction of a reducing agent. The authors highlight the novelty of their findings, emphasizing that this is the first description of fibril formation where the intricate transition into amyloids is dictated solely by redox reactions and a solitary regulatory disulfide bond.

This revised manuscript includes data confirming that p16 fibrils possess a β -sheet, amyloid structure, as demonstrated through X-ray diffraction of oxidized p16 fibers. The distinct reflection at 4.6 Å and a more diffuse reflection around 9.9 Å are consistent with the cross- β structure typical of amyloids. In my view, this result, along with ThT binding, electron microscopy, and FTIR, provides compelling evidence of an amyloid β -sheet structure.

The findings presented in the paper are novel and pave new avenues of understanding in the field. The authors properly revised the manuscript according to the comments and criticisms of the reviewers.